https://doi.org/10.1038/s41467-021-25142-0　　**OPEN**

# Metabolome and proteome analyses reveal transcriptional misregulation in glycolysis of engineered *E. coli*

Chun-Ying Wang[1,2,3], Martin Lempp[1,3], Niklas Farke[1,2], Stefano Donati [ID] [1], Timo Glatter[1] & Hannes Link [ID] [1,2✉]

Synthetic metabolic pathways are a burden for engineered bacteria, but the underlying mechanisms often remain elusive. Here we show that the misregulated activity of the transcription factor Cra is responsible for the growth burden of glycerol overproducing *E. coli*. Glycerol production decreases the concentration of fructose-1,6-bisphoshate (FBP), which then activates Cra resulting in the downregulation of glycolytic enzymes and upregulation of gluconeogenesis enzymes. Because cells grow on glucose, the improper activation of gluconeogenesis and the concomitant inhibition of glycolysis likely impairs growth at higher induction of the glycerol pathway. We solve this misregulation by engineering a Cra-binding site in the promoter controlling the expression of the rate limiting enzyme of the glycerol pathway to maintain FBP levels sufficiently high. We show the broad applicability of this approach by engineering Cra-dependent regulation into a set of constitutive and inducible promoters, and use one of them to overproduce carotenoids in *E. coli*.

[1] Max Planck Institute for Terrestrial Microbiology, Marburg, Germany. [2] Present address: Interfaculty Institute for Microbiology and Infection Medicine Tübingen, University of Tübingen, Tübingen, Germany. [3] These authors contributed equally: Chun-Ying Wang, Martin Lempp. ✉email: hannes.link@synmikro.mpi-marburg.mpg.de

Engineering synthetic metabolic pathways by inserting new enzymes into the metabolic network of microbes is a common approach to expand the spectrum of chemicals that they can overproduce[1], or gain access to new feedstocks like atmospheric $CO_2$[2]. However, synthetic metabolism interferes with the endogenous one, often in a way that impairs the cellular growth and fitness of the host. For example, overproduction pathways consume metabolites that are no longer available for the growth and metabolism of the host. This competition between synthetic and endogenous metabolism leads to a metabolic burden that causes stress responses and physiological changes of the host[3]. Eventually, metabolic burden and the accompanying perturbations to metabolism reduce the overall fitness and productivity of the engineered microbes. Therefore, the current challenge in metabolic engineering is to minimize metabolic burden, while maximizing flux through synthetic metabolic pathways.

An approach to avoid metabolic burden is to express synthetic pathways in non-growing microbes using two-stage bioprocesses[4,5]. Non-growing microbes are less susceptible to metabolic burden, because they have a lower requirement for biomass building blocks and energy. However, unlike actively growing cells, non-growing cells have a low overall metabolic activity[6], and this can limit the flux and productivity of synthetic metabolic pathways. Thus, the higher metabolic activity of growing cells is undoubtedly an advantage but requires the optimization of the enzyme levels in the synthetic metabolic pathway in such a way that sufficient resources remain for cell growth[7]. Optimal control of enzyme expression has been achieved at various levels of transcription and translation, for example by engineering promoters[8] or ribosome-binding sites[9]. However, these methods are static because they do not allow adjusting the enzyme levels to the changing internal and external conditions[10]. To dynamically control the expression levels of enzymes, feedback mechanisms have been introduced into synthetic metabolic pathways. An approach to do so is to express enzymes in the synthetic pathway under the control of promoters that bind transcription factors (TFs). The activity of the TFs in turn is controlled by intermediates or precursors of the synthetic pathway. The resulting feedback between metabolism and gene expression improved overproduction of lycopene[11], fatty acids[12,13], and precursors of isoprenoids[14]. Another approach to engineer metabolic feedback regulation of gene expression is to combine CRISPR interference with transcriptional regulators that sense stress of the host[15].

Here, we used glycerol production in *E. coli* as an example to systematically study the cause and consequences of metabolic burden in engineered bacteria. We chose the glycerol pathway because it is a simple two-step pathway that drains precursors from one of the most central pathways, glycolysis. According to the United States Department of Energy (DOE), glycerol belongs to the top ten value-added chemicals from biomass[16], and can be a precursor for other bio-based products like acrylic acid[17] and 1,3-propanediol[18]. First, we controlled the glycerol pathway with an arabinose-inducible pBAD promoter and observed that already low levels of inducer caused a growth burden. Metabolomics and proteomics data indicated that the growth burden was caused by a transcriptional response in glycolysis, notably the activation of gluconeogenesis by the transcription factor Cra. Next, we combined theoretical and experimental analysis to show that insertion of a Cra-binding site into the pBAD promoter enables higher growth rates at higher glycerol production rates. Finally, we show that this approach is generally applicable to synthetic pathways that utilize glycolytic metabolites such as carotenoid production.

## Results

**Glycerol production causes a growth burden in *E. coli*.** To investigate how induction of a synthetic metabolic pathway impacts the metabolism of the host, we expressed the glycerol biosynthesis pathway from yeast in *E. coli* (Fig. 1a). The glycerol pathway is a two-step pathway that starts from the glycolytic metabolite dihydroxyacetone phosphate (DHAP). The first reaction is catalyzed by the glycerol-3-phosphate dehydrogenase 1 (GPD1), which converts DHAP into glycerol-3-phosphate (glycerol-P). The second reaction is catalyzed by the glycerol-3-phosphate phosphohydrolase 2 (GPP2) and leads to dephosphorylation of glycerol-P into the product glycerol. Our *E. coli* strain for glycerol production expressed the two genes encoding *gpd1* and *gpp2* from a plasmid, and lacked the glycerol kinase gene (*glpK*) to prevent that glycerol is re-utilized as a carbon source (Supplementary Fig. 1). In the following, we will refer to this strain as the base strain.

We sought to control glycerol production by expressing the first enzyme in the glycerol pathway (GDP1) with an arabinose-inducible pBAD promoter, and the second enzyme (GPP2) with a strong constitutive promoter (Fig. 1a). Expressing GFP with the pBAD promoter showed a linear relationship between the concentration of arabinose (ara) and promoter activity (Fig. 1b). Thus, we expected that the pBAD promoter would allow us to linearly control the abundance of GPD1 and thereby gradually increase glycerol production (Fig. 1c). However, already low ara levels (0.3%) caused a strong growth defect and low biomass concentrations and titers of glycerol (Fig. 1d). The maximal glycerol titers were achieved with 0.1% ara (17.71 mM, Fig. 1d).

We then examined the mechanisms that caused the growth burden at higher ara levels. We excluded that the protein cost of GPD1 expression was burdensome, because expressing GFP from the pBAD promoter did not affect growth (Supplementary Fig. 2). Thus, the growth burden was likely caused by the competition between the glycerol and glycolytic flux. Flux balance analysis (FBA) with a genome-scale model of *E. coli* metabolism[19] predicted that growth and glycerol production rates follow a linear relationship (line in Fig. 1e), which reflects the trade-off between utilizing glucose for production of either biomass or glycerol. To test if the base strain followed this theoretical trade-off, we measured glycerol production rates and growth rates at three induction levels: 0, 0.1, and 0.5% (dots in Fig. 1e and Supplementary Fig. 3). However, the experimentally determined rates did not follow the theoretical trade-off that was predicted by FBA (Fig. 1e). The measured glycerol production rates and growth rates at 0.5% ara were markedly lower than the theoretical ones, thus indicating that other factors than flux balances were responsible for the growth burden.

In summary, the pBAD promoter enabled us to linearly increase protein expression (Fig. 1b). However, we could not use the pBAD promoter to modulate growth rates and glycerol production rates according to a theoretical trade-off estimated by flux balance analysis (line in Fig. 1e). Instead, at 0.5% induction of the glycerol pathway, the measured growth rates decreased much stronger than predicted by FBA (dots in Fig. 1e).

**Glycerol production activates the transcription factor Cra by decreasing fructose-1,6-bisphosphate levels.** To understand the molecular mechanisms that caused the growth burden in the base strain, we measured the metabolome at three induction levels: 0, 0.1, and 0.5% ara. Therefore, we cultured the strain in shake flasks and collected samples for metabolomics by fast filtration (Fig. 2a). The metabolome data covered 96 metabolites (Supplementary Fig. 4) that remained relatively constant at 0.1% ara but displayed

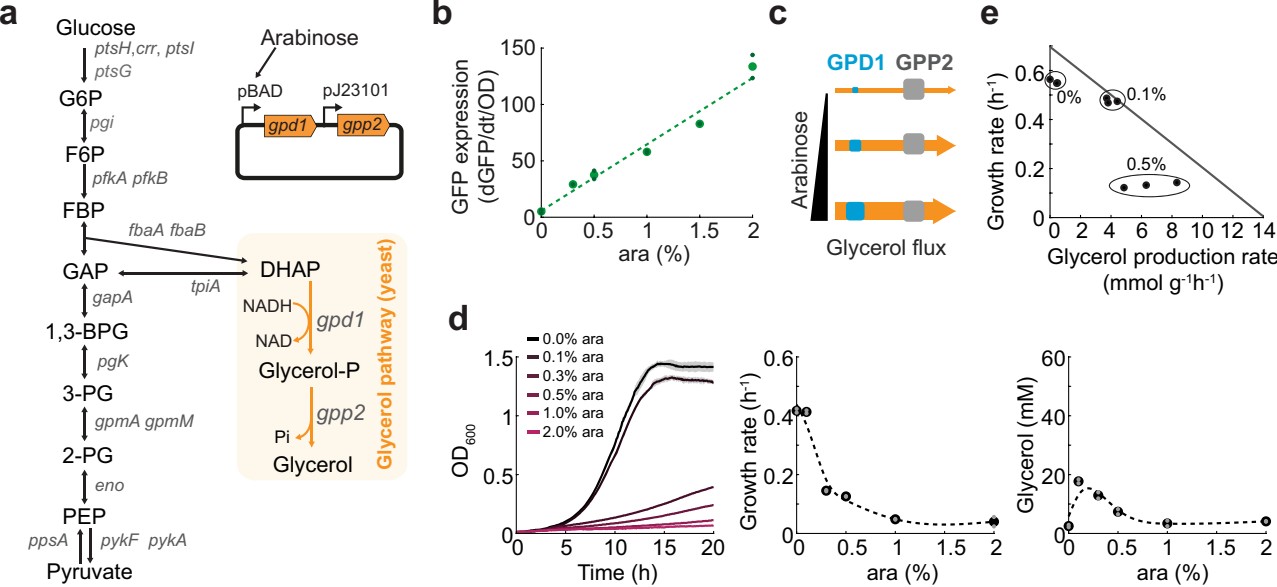

**Fig. 1 Overproduction of glycerol causes a growth burden in *E. coli*. a** Metabolic map of *E. coli* glycolysis and the synthetic glycerol pathway (orange). The synthetic pathway consists of two enzymes from *S. cerevisiae*: glycerol-3-phosphate dehydrogenase 1 (GPD1) and glycerol-3-phosphate phosphohydrolase 2 (GPP2). The genes encoding the two enzymes (*gpd1* and *gpp2*) were expressed from a plasmid using the arabinose-inducible pBAD promoter for *gpd1* and the constitutive promoter pJ23101 for *gpp2*. **b** Activity of the pBAD promoter at different arabinose levels. GFP fluorescence and $OD_{600}$ were measured in $n = 2$ plate reader cultures, and promoter activity was calculated as dGFP/dt/OD by regression analysis between 7 and 9 h. **c** Schematic of the control strategy for the synthetic glycerol pathway. GPP2 is expressed in excess to ensure that GPD1 is the rate-limiting step. GPD1 levels are varied by inducing the pBAD promoter with different amounts of arabinose. Size of boxes indicates enzyme levels, size of arrows indicate flux through the pathway. **d** *gpd1* and *gpp2* were expressed from a plasmid in an *E. coli* strain lacking *glpK* (base strain). The base strain was cultured in 96-well plates. Growth was measured in a plate reader at different induction levels of GPD1 (0, 0.1, 0.3, 0.5, 1, and 2% ara). Glycerol in the medium was measured after 24 h. Growth rates were determined by regression analysis between 5 and 10 h. Growth curves and dots show the means of $n = 2$ plate reader cultures. **e** Theoretical relationship between glycerol flux and growth rate based on flux balance analysis with a genome-scale model of *E. coli* metabolism (*i*ML1515). Dots are growth rates and glycerol production rates measured in shake flask cultures of the base strain at 0, 0.1, and 0.5% ara. Source data are provided in the Source Data file.

strong changes at 0.5% ara (Fig. 2b). The most strongly decreased metabolite at 0.5% ara was the direct precursor for the glycerol pathway, DHAP (Fig. 2b and Supplementary Fig. 4). Also, fructose-1,6-bisphosphate (FBP), which is directly upstream of DHAP, was one of the most strongly responding metabolites and decreased more than 5-fold in the presence of 0.5% ara (Fig. 2b). These data demonstrate that glycerol overproduction perturbs metabolites near the entry point of the engineered pathway.

FBP is a regulatory metabolite that is responsible for a glycolytic flux-dependent regulation of gene expression in *E. coli*[20]. FBP inhibits the activity of the transcription factor Cra, which inhibits the expression of genes encoding glycolytic enzymes and activates gluconeogenesis-related genes (Fig. 2c). Although it is currently unclear whether FBP is a direct or indirect effector of Cra[21], it is widely assumed that the concentration of FBP affects Cra activity. Correspondingly, we wondered whether the low concentration of FBP (at 0.5% ara) activated Cra and thereby changed gene expression and enzyme levels. To test this, we probed the proteome at the three induction levels, and inspected the abundance of a total of 38 enzymes in glycolysis and gluconeogenesis (Fig. 2d). Similar to metabolites, enzyme levels changed stronger at 0.5% induction than at 0.1% induction. To test if proteome changes were caused by Cra, we measured the proteome of the Cra deletion strain (Δ*cra*) as a reference (Fig. 2d). One of the most strongly decreased enzymes in the Δ*cra* strain was the phosphoenolpyruvate synthetase (PpsA). The strong effect of Cra on the expression of PpsA is consistent with previous studies, which showed that the *ppsA* promoter is under the direct control of Cra[22]. In our base strain, PpsA was one of the most strongly increased enzymes at 0.5% ara

(Fig. 2d), thus indicating a high activity of Cra in this strain. Moreover, the base strain had low levels of glycolytic enzymes that are known to be repressed by Cra, such as glyceraldehyde-3-phosphate dehydrogenase (GapA).

Taken together, proteome and metabolome data suggest that induction of the glycerol pathway with 0.5% ara decreased the concentration of FBP. This, in turn, activated the transcription factor Cra which then downregulated enzymes in glycolysis (e.g., GapA) and upregulated enzymes in gluconeogenesis (e.g., PpsA). Because cells grew on a glucose minimal medium we hypothesized that activation of gluconeogenesis was responsible for the growth burden. We confirmed this hypothesis by deleting *cra* in the base strain (Fig. 2e). The resulting Δ*cra* strain grew indeed better than the base strain at high induction of the synthetic glycerol pathway, and the maximum glycerol titers increased 1.6-fold (compare Figs. 2e and 1d). Thus, Cra-regulation contributes to the growth burden of glycerol overproduction in *E. coli*.

**A metabolic model predicts optimization strategies for glycerol production**. To obtain additional evidence that transcriptional regulation by Cra is a problem for glycerol production, we developed a small kinetic model (Fig. 3a). The model included one metabolite (FBP) and two enzymes e1 and e2. Enzyme e1 corresponds to glyceraldehyde-3-phosphate dehydrogenase (GapA) in lower glycolysis and e2 is GPD1 in the glycerol pathway. FBP influenced reaction rates in lower glycolysis ($r_{lower\_glycolysis}$) and in the glycerol pathway ($r_{glycerol}$) according to Michaelis–Menten kinetics, which are a well-established kinetic format for enzymatic reactions[23]. Similar to flux balance analysis

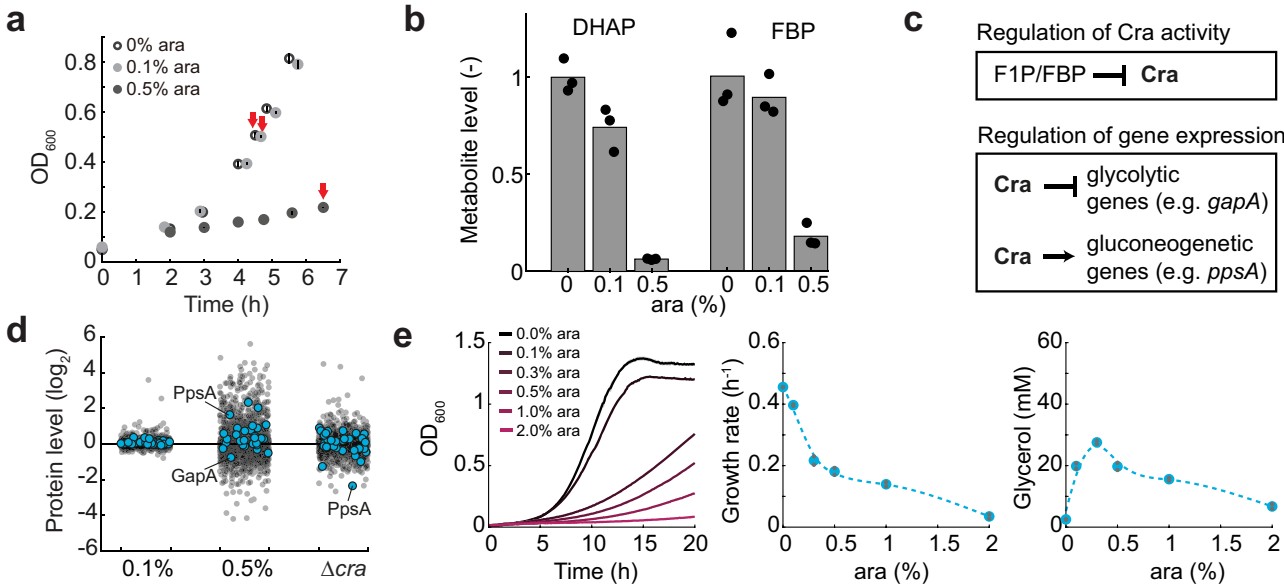

**Fig. 2 Induction of the glycerol pathway activates the transcription factor Cra. a** Growth of the base strain (*E. coli* Δ*glpK* with the pBAD glycerol plasmid) at different induction levels (0, 0.1, and 0.5% ara). The base strain was cultured in $n = 3$ shake flasks and samples for metabolomics and proteomics were collected at the time points indicated by red arrows. Dots are means and lines are standard deviations of the $OD_{600}$ in $n = 3$ shake flask cultures. **b** Intracellular concentration of dihydroxyacetone phosphate (DHAP) and fructose-1,6-bisphosphate (FBP). Data are normalized to the 0% culture. Dots are data of $n = 3$ independent shake flask cultures and bars are the mean. **c** Fructose-1-phosphate (F1P) and FBP inhibit the activity of the transcription factor Cra. Cra activates the expression of genes encoding gluconeogenic enzymes (e.g., *ppsA*) and represses those of glycolytic enzymes. **d** Proteome data showing the relative abundance of proteins in the base strain with 0.1% ara and 0.5% ara. Data are normalized to the base strain with 0% ara. Δ*cra* is the proteome of a *cra* deletion strain, normalized to the wild-type strain. Dots are means of samples from $n = 3$ independent shake flask cultures (**a**). Shown are only protein levels with a relative standard deviation smaller than 20%. Blue dots are enzymes that belong to glycolysis or gluconeogenesis in the *i*ML1515 model. **e** The glycerol pathway was expressed in the Δ*cra* strain. The Δ*cra* strain was cultured in 96-well plates. Growth was measured in a plate reader at different induction levels of GPD1 (0, 0.1, 0.3, 0.5, 1, and 2% ara). Glycerol in the medium was measured after 24 h. Growth rates were determined by regression analysis between 5 and 10 h. Growth curves and dots show the means of $n = 2$ plate reader cultures. Source data are provided in the Source Data file.

**Table 1 Values and units of parameters and variables.**

| Parameter/species | Lower bound | Upper bound | Unit |
|---|---|---|---|
| $r_{upper\_glycolysis}$ | 40.87 | 40.87 | mM min$^{-1}$ |
| $k_{cat,1}$ | Calculated | | min$^{-1}$ |
| $K_{m,1}$ | 0.01 | 10 | mM |
| $k_{cat,2}$ | 511.5 | 5115 | min$^{-1}$ |
| $K_{m,2}$ | 0.01 | 10 | mM |
| $\beta_{1,max}$ | Calculated | | mM min$^{-1}$ |
| $\beta_{2,max}$ | 0.0017 | 0.0017 | mM min$^{-1}$ |
| ind | 0 | 1 | – |
| α1 | 1 | 2 | – |
| α2 | 1 | 2 | – |
| $\mu_{initial}$ | 0.01 | 0.01 | min$^{-1}$ |
| FBP | 1 | 1 | mM |
| e1(GapA) | 0.0238 | 0.0238 | mM |
| e2(GPD1) | 0 | 0 | mM |

(Fig. 1e), we assumed a constant influx in the model and fixed the reaction rate in upper glycolysis to 4.9 mmol g$^{-1}$ h$^{-1}$. This means that FBP is produced at a constant rate and it is either used for glycerol production or for growth according to the mass balances in Eq. (1).

In total, we analyzed three different models, each with a different regulatory structure (Fig. 3b). A model of the base strain included an interaction, in which FBP activates the expression of enzyme e1. This interaction resembled transcriptional regulation of lower glycolysis by Cra. A second model of the Δ*cra* strain had no regulation. In a third model, FBP activates the expression of both enzymes e1 and e2. In this doubly regulated model (2x*cra* model) both lower glycolysis and the glycerol pathway were subject to Cra-regulation. We simulated Cra-regulation with a power-law term that affects the maximal enzyme expression rate. Since the power-law term equals one in the un-induced state, all models share the same parameter set.

The three models were analyzed with 5000 parameter sets that were randomly sampled from physiologically meaningful ranges based on literature values (Table 1). We sampled the power-law exponent between 1 and 2, in order to ensure that Cra-regulation depends at least linearly on the concentration of FBP and to avoid instabilities that can occur at exponents <2. For each of the 5000 parameter sets, we calculated the maximal glycerol production rate ($r_{glycerol,MAX}$) that can possibly be achieved given the specific set of parameters. To estimate $r_{glycerol,MAX}$, we made use of a numerical continuation method[24], which iteratively increases the expression rate of enzyme e2 ($\beta_2$) and computes the new steady state for FBP, e1, and e2. After each iteration, the continuation method determines the stability of the model by inspecting the eigenvalues of the Jacobian matrix[24], and terminates if instabilities occur in the model. If the model remains stable, the continuation method terminates at the maximal expression rate of e2 ($\beta_{2,max}$), which we defined as the rate were 20% of the ribosomes translate e2. Thus, $r_{glycerol,MAX}$ is the glycerol production rate at the termination point of the continuation method and we obtained 5000 values of $r_{glycerol,MAX}$ for each of the three models (Fig. 3c).

The distribution of the 5000 $r_{glycerol,MAX}$ values showed that the Δ*cra* model performed better than the base strain model, because more parameter sets achieved higher maximal glycerol

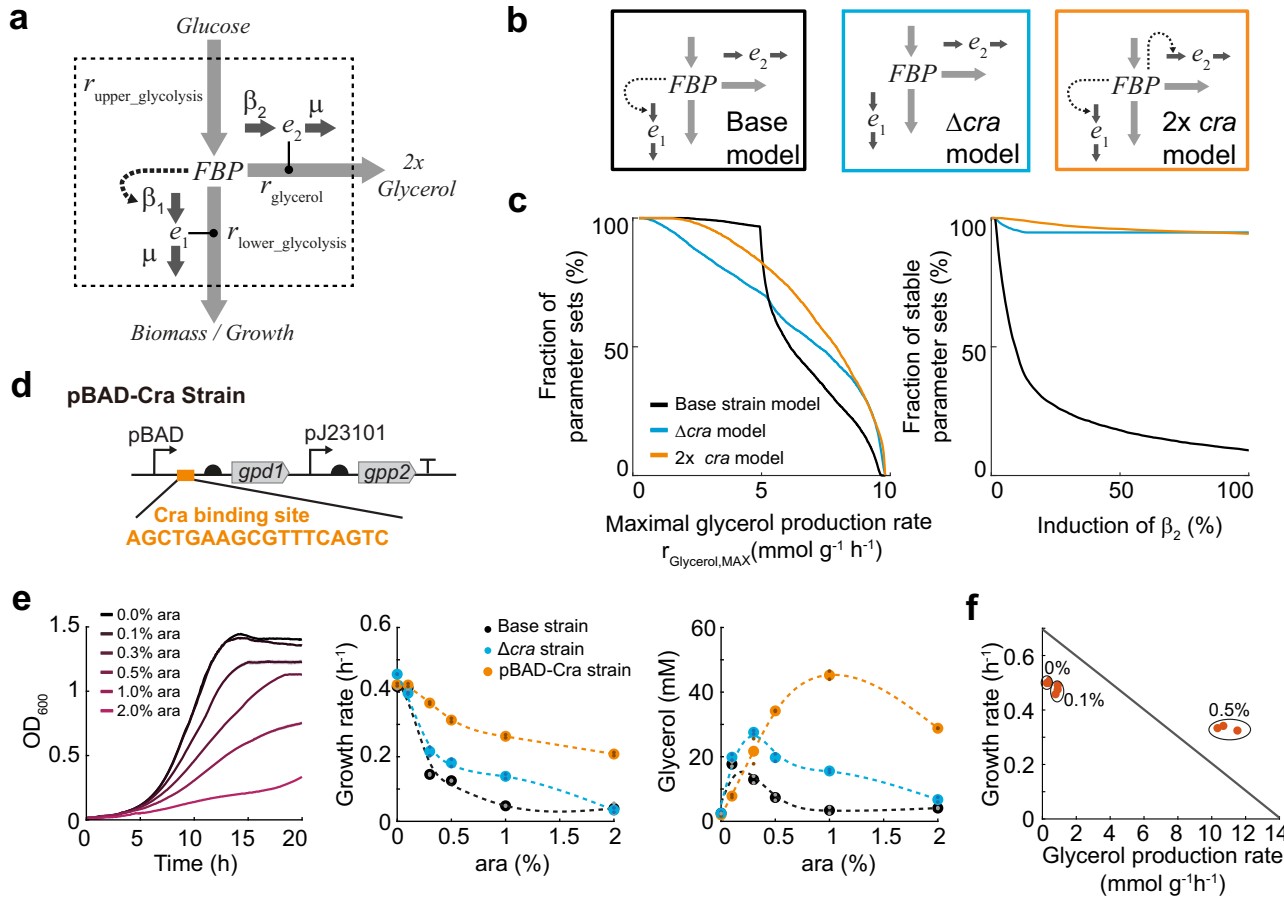

**Fig. 3 Theoretical and experimental analysis of glycerol production with a Cra-regulated pBAD promoter. a** Stoichiometry and structure of the kinetic model. The dashed box is the model boundary. FBP is a metabolite in glycolysis, and e1 and e2 are enzymes. Solid, gray arrows are metabolic reactions. The input reaction in upper glycolysis is fixed to 4.9 mmol g$^{-1}$ h$^{-1}$. The reaction in lower glycolysis and the glycerol pathway depends on FBP levels and enzyme levels, according to Michaelis–Menten kinetics. The dashed arrow is FBP-activation of $\beta_1$ (the expression rate of e1) and represents the net effect of Cra regulation: FBP inhibits Cra; Cra inhibits $\beta_1$. Gradual increases of $\beta_2$ (the expression rate of e2) simulate induction with arabinose. **b** A model of the base strain (black), a model of the $\Delta cra$ strain (blue), and a model with additional Cra regulation of e2 expression (orange). **c** The three models in **b** were simulated with the same 5000 parameter sets that were obtained by random sampling (see also Table 1). For each parameter set, $\beta_2$ (the expression rate of e2) was increased until the model became unstable or until the expression rate $\beta_2$ reached the maximum. Shown is the maximal glycerol flux that was achieved with each model as a cumulative sum distribution. Robustness is shown as the percentage of the 5000 models that remain stable at a given induction level. **d** The pBAD promoter was engineered by inserting a Cra-binding site between the promoter and the ribosome-binding site, resulting in the pBAD-Cra promoter. Cra-FBP regulation should ensure that low FBP levels repress the pBAD-Cra promoter. The pBAD promoter in the base strain (Fig. 1a) was replaced with the pBAD-Cra promoter, resulting in the pBAD-Cra strain. **e** The pBAD-Cra strain (orange) was cultured in 96-well plates. Growth was measured in a plate reader at different induction levels of GPD1 (0, 0.1, 0.3, 0.5, 1, and 2% ara). Glycerol in the medium was measured after 24 h. Growth rates were determined by regression analysis between 5 and 10 h. Small dots show data from $n = 2$ plate reader cultures and big dots are the mean. Data of the base strain (black) and the $\Delta cra$ strain (blue) are shown as a reference (same data as in Figs. 1d and 2e). **f** Growth rates and glycerol production rates of the pBAD-Cra strain measured in shake flasks, at 0, 0.1, and 0.5% ara. The line is the theoretical relationship between glycerol flux and growth rate shown in Fig. 1e. Source data are provided in the Source Data file.

production rates ($r_{glycerol,MAX}$) with the $\Delta cra$ model than with the base strain model (Fig. 3c). This matched the experimental observation that the $\Delta cra$ strain performed better than the base strain. The underlying assumption was: the more parameter sets achieve high glycerol fluxes, the higher the likelihood that the real system would achieve them too. The model of the base strain did not achieve high glycerol production rates, because the model was not stable at higher induction levels (Fig. 3c). To better understand the origin of these instabilities, we performed time-course simulations with the three models and an average parameter set (Supplementary Fig. 5). The time-course simulations matched the results obtained with the continuation method, thus confirming that both numerical approaches yield the same results. We simulated the models at different induction levels and

the base model was not stable at higher induction, because enzyme e2 increased exponentially. Thus, there is a critical point where the expression rate of e2 exceeds its dilution by growth. These imbalances are probably amplified by Cra-regulation, because Cra downregulates $e1$ and thus growth.

The $\Delta cra$ model, in contrast, was stable at almost all induction levels. The best model in our analysis was the 2x$cra$ model. With this model, the highest fraction of parameter sets achieved high glycerol fluxes (Fig. 3c). Further, the stability of the 2x$cra$ model was similar to the $\Delta cra$ model (Fig. 3c). Thus, the model analysis predicted that engineering Cra-regulation into the glycerol pathway should lead to higher glycerol production rates and we next tested this prediction experimentally.

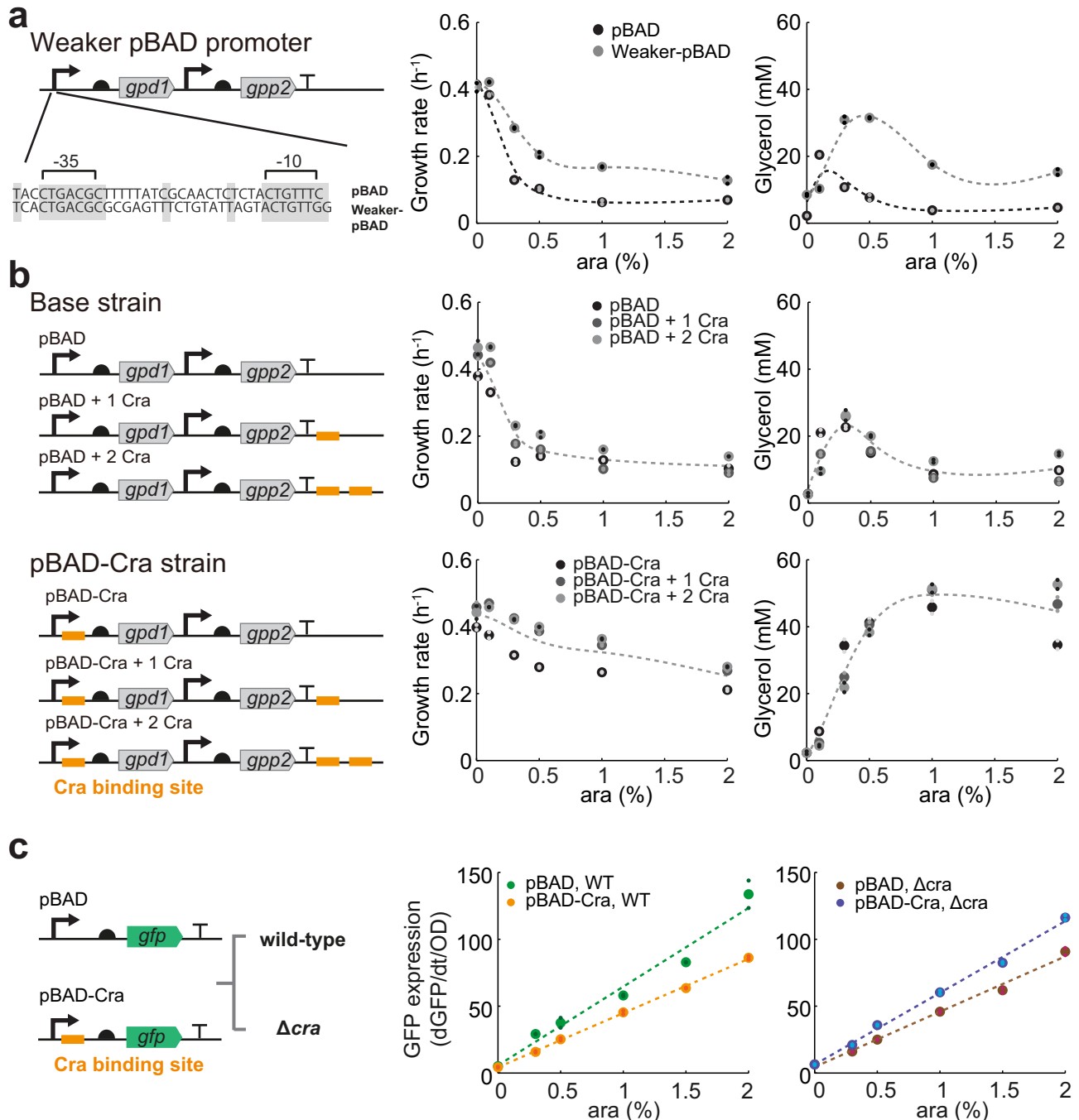

**Fig. 4 Alternative strategies to engineer the pBAD promoter. a** The pBAD promoter was mutated between the −35 and −10 boxes[25]. The pBAD promoter in the base strain (Fig. 1a) was replaced with the mutated pBAD promoter and cultured in 96-well plates. Growth was measured in a plate reader at different induction levels of GPD1 (0, 0.1, 0.3, 0.5, 1, and 2% ara). Glycerol in the medium was measured after 24 h. Growth rates were determined by regression analysis between 5 and 10 h. Small dots show data from $n = 2$ plate reader cultures and big dots are the mean. **b** Same as **a**, for strains expressing *gpd1* with engineered pBAD promoters that have 0 to 3 Cra-binding sites inserted at different positions. The position of Cra-binding sites is indicated in orange. **c** GFP plasmids with or without Cra-binding site were expressed in the wild-type (WT) or the Δ*cra* strain. ($n = 2$, pBAD, WT is the same as in Fig. 1b). Source data are provided in the Source Data file.

**A Cra-regulated pBAD promoter improves growth and glycerol production.** To engineer a pBAD promoter that is repressed by Cra, we inserted the consensus binding sequence of Cra between the promoter region and the ribosome-binding site (Fig. 3d). Then we expressed GPD1 under the control of this pBAD-Cra promoter and introduced the plasmid in *E.coli* Δ*glpK* to create the pBAD-Cra strain. This strain grew indeed much better than the base strain and maintained growth even at full

induction with 2% arabinose (orange one in Fig. 3e). At 0.5% ara, growth and glycerol production rates of the pBAD-Cra strain were even higher than the rates at the theoretical trade-off frontier (Fig. 3f and Supplementary Fig. 6). These data confirm the model prediction that a doubly Cra-regulated strain performs better than the base strain and the Δ*cra* strain.

Next, we compared the activities of the pBAD promoter and the pBAD-Cra promoter in the context of our production system.

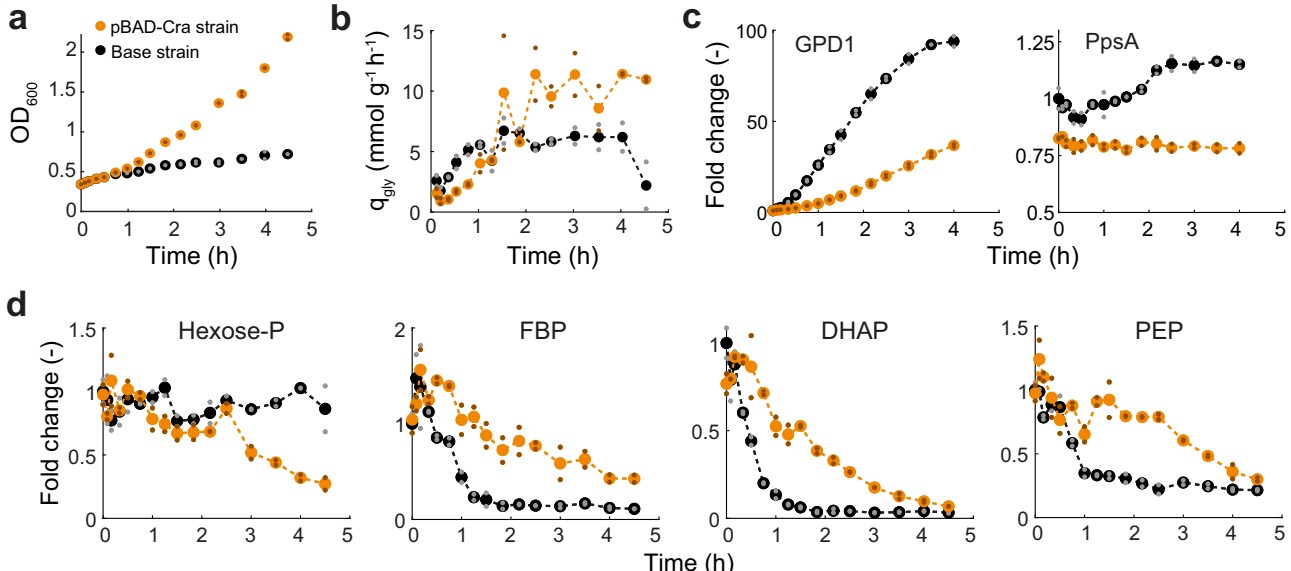

**Fig. 5 Dynamic metabolite and protein responses to induction of the glycerol pathway. a** Growth of the base strain (black) and the pBAD-Cra strain (orange). Glycerol production was induced by supplementing 0.5% ara to the medium at $t = 0$ h. Small dots show $n = 2$ independent shake flasks, and large dots are the mean. **b** Specific glycerol production rates (mmol $g^{-1}$ $h^{-1}$) were calculated for two adjacent time points as $\Delta c_{gly}/c_x/\Delta t$, where $\Delta t$ is the time interval between the two time points; $\Delta c_{gly}$ is the difference of glycerol in the medium; $c_x$ is the biomass concentration. **c** Time-course of GPD1 levels and phosphoenolpyruvate synthetase (PpsA) levels. Proteomics samples were collected from the base strain (black) and the pBAD-Cra strain (orange) (**a**). Data are normalized to the 0 h time point of the base strain. **d** Time-course of the concentration of intracellular metabolites (Hexose-P hexose-phosphates, FBP fructose-1,6-bisphosphate, DHAP dihydroxyacetone phosphate, PEP phosphoenolpyruvic acid). Metabolomics samples were collected from the base strain (black) and the pBAD-Cra strain (orange) (**a**). Data are normalized to the 0 h time point of the base strain. Note that in **b**–**d**, the small dots show samples from $n = 2$ independent shake flasks and big dots are the mean. Source data are provided in the Source Data file.

Therefore, we replaced GPD1 with a GPD1-GFP fusion protein (Supplementary Fig. 7). The pBAD-Cra promoter had a 3.7-fold lower activity than the pBAD promoter at 0.5% ara, showing that insertion of the Cra-binding site reduced GPD1 expression. Flow cytometry data revealed that the cell-to-cell variation in GPD1-GFP content was independent of the promoter (Supplementary Fig. 8), indicating that all the cells of the pBAD-Cra population had a lower promoter activity.

Thus, the pBAD-Cra promoter is a 3.7-fold weaker version of the pBAD promoter, probably because Cra represses the promoter. In principle, the same effect could be achieved by other mutations that decrease the activity of the pBAD promoter, for example, mutations between the −10 and −35 boxes[25]. Therefore, we constructed a pBAD promoter with mutations between the −10 and −35 boxes that decrease activity by a factor of two[25], and analyzed the resulting pBAD-weak strain. The pBAD-weak strain grew indeed better than the base strain and achieved higher glycerol titers (Fig. 4a). However, the pBAD-weak strain performed worse than the pBAD-Cra strain, which might be due to the different activities of the pBAD-Cra and the pBAD-weak promoter (3.7-fold and 2-fold lower activity than the original pBAD promoter, respectively). Another explanation for the better growth of the pBAD-Cra strain is that the pBAD-Cra promoter is dynamic and the pBAD-weak promoter is static.

If Cra actively inhibits the pBAD-Cra promoter, we expected that insertion of Cra-binding sites outside of the promoter region would have no effect. Indeed, inserting 1 to 3 Cra-binding sites improved growth only when a binding site was inserted directly after the pBAD promoter (Fig. 4b). Even two Cra-binding sites outside of the promoter region gave no improvement. This demonstrates that Cra actively inhibits the pBAD-Cra promoter, and that the improvements are not a consequence of titrating Cra away from its genomic targets.

To obtain further evidence that the pBAD-Cra promoter is functional, we measured the activity of the pBAD promoter and the pBAD-Cra promoter with GFP, both in the wild-type and the Δcra strain. In the wild-type, the pBAD-Cra promoter had lower activity than the pBAD promoter (Fig. 4c), thus indicating that Cra inhibits the promoter. In the Δcra strain, however, the pBAD-Cra promoter had slightly higher activity than the pBAD promoter (Fig. 4c). These results show that the pBAD-Cra promoter is functional: Cra inhibits the pBAD-Cra promoter and this regulation is missing in the absence of Cra (in the Δcra strain). Thus the lower activity of the pBAD-Cra is not merely due to sequence changes but due to active inhibition by Cra.

To further demonstrate the broad utility of this approach we inserted Cra into a constitutive promoter and the pTet promoter (Supplementary Fig. 9). In both cases, the strain with the Cra-regulated promoter variant grew better than the strain with the original version. This suggests that Cra inhibits these promoters and automatically reduces their activity.

**The Cra-regulated pBAD promoter maintains high FBP levels at high glycerol fluxes**. To better understand the dynamic nature of Cra-regulation, we measured the dynamic changes in the metabolome and proteome upon induction of the glycerol pathway. Therefore, we induced the base strain and the pBAD-Cra strain with 0.5% ara and collected metabolomics and proteomics samples for the subsequent 4.5 h. Additionally, we measured growth (Fig. 5a) and the concentration of glycerol in the medium in order to calculate the flux through the glycerol pathway (Fig. 5b). The pBAD-Cra strain grew again much better than the base strain (Fig. 5a). The growth defect of the base strain appeared 1 h after inducer addition, but at this time point, both strains had similar glycerol production rates (~5 mmol $g^{-1}$ $h^{-1}$, Fig. 5b). After 2 h, the glycerol production rate was even higher in the pBAD-Cra strain (10 mmol $g^{-1}$ $h^{-1}$) than in the base strain

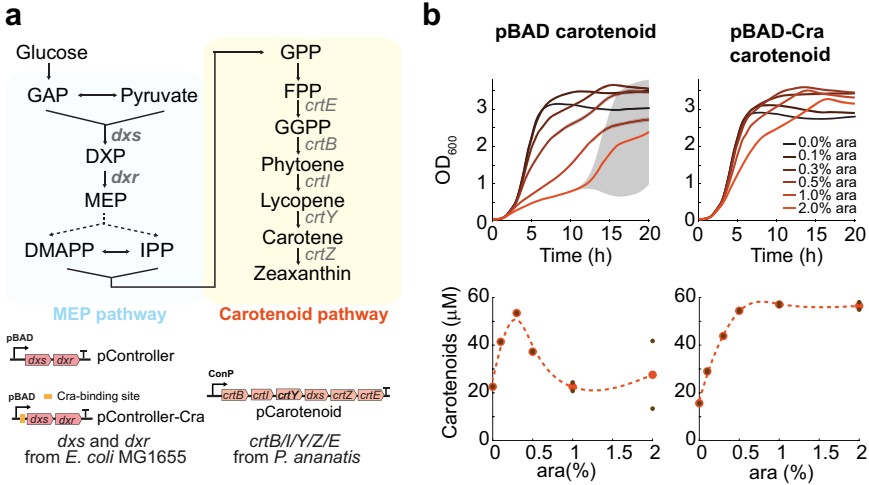

**Fig. 6 A Cra-regulated pBAD promoter improves overproduction of carotenoids. a** Metabolic map of the *E. coli* methylerythritol phosphate (MEP) pathway and the synthetic carotenoid pathway. Expression of the first enzymes in the MEP pathway (1-deoxy-D-xylulose-5-phosphate synthase, Dxs; and 1-deoxy-D-xylulose 5-phosphate reductoisomerase, Dxr) was controlled with a pBAD promoter (plasmid pController) or with a pBAD-Cra promoter (plasmid pController-Cra). The carotenoid pathway consists of five enzymes from *P. ananatis*: geranylgeranyl diphosphate synthase (CrtE), phytoene synthase (CrtB), phytoene dehydrogenase (CrtI), lycopene beta-cyclase (CrtY), and beta-carotene hydroxylase (CrtZ). These five genes were expressed under the control of a native constitutive promoter from *P. ananatis* and one additional *dxs* gene from *E. coli* (plasmid pCarotenoid). **b** Plasmids pCarotenoid and pController were co-expressed in *E. coli* (pBAD carotenoid strain). Both strains were grown in a plate reader at different induction levels for Dxs and Dxr (0, 0.1, 0.3, 0.5, 1, and 2% ara). The total carotenoid content was measured after 24 h. Growth curves and dots show means of $n = 2$ cultures, and gray shaded areas show the difference between the two cultures. GAP glyceraldehyde-3-phosphate, DXP 1-deoxy-D-xylulose-5-phosphate, MEP 2-C-methyl-D-erythritol-4-phosphate, DMAPP dimethylallyl diphosphate, IPP isopentenyl diphosphate, GPP geranyl diphosphate, FPP farnesyl diphosphate, GGPP geranylgeranyl diphosphate. Source data are provided in the Source Data file.

(6 mmol g$^{-1}$ h$^{-1}$). This suggested that it was not glycerol production per se, which impaired growth of the base strain, but rather other effects such as the higher expression levels of GPD1 (Fig. 5c).

We then hypothesized that after 1 h Cra was activated in the base strain, while Cra should be less active in the pBAD-Cra strain. To test this hypothesis, we used again the abundance of PpsA as a proxy for Cra activity. In the base strain, PpsA levels increased 1 h after inducer addition, which matches the time when this strain shows a growth defect (Fig. 5c). In the pBAD-Cra strain, however, PpsA levels remained constant after induction, suggesting that in this strain the activity of Cra remains below a threshold that activates gluconeogenesis.

If Cra is less active in the pBAD-Cra strain than in the base, we expected that the latter has lower FBP levels. Indeed, FBP decreased rapidly after inducing the base strain (Fig. 5d). The pBAD-Cra strain, in contrast, had always higher FBP levels than the base strain despite its higher flux in the glycerol pathway. The concentration of other glycolytic metabolites (hexose-P, DHAP, and PEP) was also higher in the pBAD-Cra strain than in the base strain (Fig. 5d). Additionally, we confirmed that the pBAD-Cra strain maintained higher concentrations of FBP under steady-state conditions, by probing the metabolome at constant induction with 0, 0.1, and 0.5% ara (Supplementary Fig. 4).

Thus, the pBAD-Cra strain can maintain higher FBP levels at higher glycerol production rates than the base strain. This suggested that the interaction between FBP and Cra, in combination with the pBAD promoter, counteracts decreases of FBP: (i) if FBP falls below a critical value, Cra activity increases, (ii) higher Cra activity represses the pBAD-Cra promoter and decreases GPD1 expression, and (iii) lower expression of GDP1 will restore the concentration of FBP. Thus our data indicate that this feedback regulation is functional, because it enabled high FBP levels and at the same time high glycerol production rates, which presumably prevented that *E. coli* switches from glycolysis to

gluconeogenesis. However, further experiments are required to experimentally investigate how this feedback loop shapes metabolism in space and time[26].

**Cra-regulation improves the growth of carotenoid overproducing *E. coli*.** Because many bio-based chemicals use glycolytic metabolites as precursors, we wondered if the Cra-regulated pBAD promoter could have broader applicability. Therefore, we used the pBAD promoter (in its original version and with Cra-regulation) to control a synthetic metabolic pathway for the overproduction of carotenoids (Fig. 6a). Biosynthesis of carotenoids starts from the glycolytic metabolites pyruvate and glyceraldehyde-3-phosphate (GAP), which are converted by the methylerythritol phosphate (MEP) pathway of *E. coli* into farnesyl diphosphate (FPP). FPP is then further converted into carotenoids by heterologous enzymes from *Pantoea ananatis* (Fig. 6a). The first two enzymes in the MEP pathway (Dxs and Dxr) were overexpressed from a plasmid using the two versions of the pBAD promoter, and we refer to the two plasmids as pController and pController-Cra, respectively. The remaining enzymes in the carotenoid pathway (CrtE/B/I/Y/Z) were expressed from a second plasmid (pCarotenoid) using a native promoter of *P. ananatis*.

Expressing only the pCarotenoid plasmid led to a basal production of carotenoids and did not influence cell growth (Supplementary Fig. 10). Carotenoid production increased almost 3-fold when *E. coli* carried the pCarotenoid plasmid together with either the pController or pController-Cra plasmid (Fig. 6b). However, with the pController plasmid, higher carotenoid levels were only achieved within a small range of inducer, whereas the pController-Cra plasmid performed well over a broader range of ara levels. Thus, similar to the glycerol pathway, we observed that the promoter with Cra regulation enables higher growth rates at high induction levels of a synthetic carotenoid pathway, and that the higher amount of inducer does not impact cell growth and

productivity (Fig. 6b). Taken together, the results suggest that the Cra-regulated pBAD promoter is generally applicable and may enable bioengineers to regulate the expression of a wide range of synthetic pathways that use glycolytic metabolites.

## Discussion

In this study, we used the arabinose-inducible pBAD promoter to control synthetic metabolic pathways in *E. coli*. While, in the case of GFP expression, the pBAD promoter showed a linear relationship between the concentration of inducer (ara) and expression rates, this was not observed for glycerol overproduction. In the latter case, the problem was that a small increase of inducer was sufficient to cause a strong growth burden that constrained productivity. A consequence of the growth burden was a low biomass concentration and, consequently, low glycerol titers at the end of the cultivations.

A classical view is that synthetic metabolic pathways are a burden for industrial microbes because they consume cellular resources[27]. Recent multi-omics data suggest that synthetic metabolic pathways can place an additional burden on cellular metabolism by perturbing the proteome and metabolome[28]. Perturbations of the metabolome could be critical especially if they alter the concentration of regulatory metabolites that control the expression and function of proteins, for example via metabolite-protein interactions[29,30]. Here, we observed that glycerol overproduction decreased the concentration of glycolytic metabolites, and that this caused misregulation at the level of transcription. More specifically, low concentrations of the regulatory metabolite FBP activated the transcription factor Cra and thereby upregulated the expression of gluconeogenesis enzymes likes PpsA. These results demonstrate the importance of maintaining regulatory metabolites above a critical threshold in engineered microbes. However, synthetic pathways often lack the regulatory mechanisms that maintain metabolite concentration homeostasis in natural metabolic pathways, such as directed overflow[31], allosteric enzyme regulation[32], or transcriptional regulation[33]. Here we show that engineering transcriptional regulation (Cra-regulation) in a synthetic glycerol pathway can help to maintain regulatory metabolites above a critical threshold: the pBAD-Cra strain has higher FBP levels and at the same time a higher glycerol flux than the base strain. This supports the hypothesis that Cra-dependent regulation counteracts a decline in the concentration of FBP by downregulating the expression of the glycerol pathway in response to decreasing FBP levels. However, as it remains an open question whether this regulation is truly dynamic, we cannot rule out the possibility that, due to the constant inhibitory activity of Cra, the pBAD-Cra promoter simply functions as a weaker pBAD promoter. Future studies should clarify whether the pBAD-Cra promoter automatically adapts to new conditions, e.g., by shifting the glycerol producers between different environments.

Previous studies mainly focused on the transcriptional consequences of promoter engineering[34,35], or demonstrated the ability of engineered promoters to increase the productivity of synthetic pathways[11–15]. Here, we combined metabolomics and proteomics to study the consequences of engineered promoters at the level of host metabolism. From a biotechnological perspective, this approach will help to engineer improved production strains that can autonomously buffer external perturbations in industrial-scale bioreactors[36,37], and internal perturbations like gene expression noise[38,39].

## Methods

**Construction of strains and plasmids.** Strains and plasmids are listed in Supplementary Table S1. Strains derived from *Escherichia coli* K-12 MG1655 (wild type, DSMZ No. 18039) were used for glycerol and GFP production. Strains derived from *E. coli* BW25113 (KEIO collection[40]) were used for carotenoid production. Consensus sequences of Cra are AGCTGAAGCGTTTCAGTC (from *epd* gene). All oligonucleotides used for cloning were synthesized by Eurofins Genomics (Germany GmbH) and are listed in Supplementary Table S2. Target genes were amplified to obtain linear fragments by PCR using Q5 High-Fidelity DNA Polymerase (M0491L, BioLabs). Circular polymerase extension cloning (CPEC)[41] and Gibson assembly (E2611S, Biolabs) were used for cloning. The No-SCAR system[42] for genome editing was obtained from Addgene (#62654, #62655, and #62656, see Supplementary Table S1) and used for the construction of ΔglpK and Δcra. The genes for glycerol-producing enzymes were cloned from yeast chromosomal DNA (*Saccharomyces cerevisiae* SEY6210) into the pBAD promoter. The Cra consensus binding sequence was inserted into the pBAD-Cra plasmid by PCR. The pController plasmid carries *dxs* and *dxr* genes from *E. coli* MG1655. The pController-Cra plasmid was derived from the pController plasmid by inserting the Cra-binding site.

**Cultivations.** All cultivations were performed using an M9 minimal medium with 5 g l$^{-1}$ glucose. M9 medium was composed of (per liter): 42.2 mM Na$_2$HPO$_4$, 22 mM KH$_2$PO$_4$, 11.3 mM (NH$_4$)$_2$SO$_4$, 8.56 mM NaCl, 1 mM MgSO$_4$·7H$_2$O, 100 μM CaCl$_2$·2H$_2$O, 60 μM FeCl$_3$, 7.6 μM CoCl$_2$·6H$_2$O, 7.1 μM MnSO$_4$·2H$_2$O, 7 μM CuCl$_2$·2H$_2$O, and 6.3 μM ZnSO$_4$·7H$_2$O. For pre-cultures, frozen bacterial stocks were plated on LB agar plates with the respective antibiotics and single colonies were used to inoculate 5 ml LB-pre-cultures in tubes. From this first pre-culture a second M9 pre-culture was inoculated 1:1000 and incubated overnight at 37 °C under shaking. For cultivations in microtiter plates, 96-well flat transparent plates (Greiner Bio-One International) containing 150 μl M9 minimal medium were inoculated 1:150 from the overnight M9-culture. Online measurements of optical density at 600 nm (OD$_{600}$) with the glycerol production strains were performed at 37 °C with shaking in a plate reader (Epoch, BioTek Instruments Inc, USA). Online measurements of OD$_{600}$ and additional measurements of GFP fluorescence (excitation 490 nm, emission 530 nm) of the GFP production strain were performed at 37 °C with shaking in a plate reader (Synergy, BioTek Instruments Inc, USA). Growth rates were calculated as dln(OD$_{600}$)/d$t$ by linear regression over the indicated time windows. For cultivations in shake flask, a 500 ml shake flask containing 35 ml M9 minimal medium (5 g l$^{-1}$ glucose) was inoculated 1:150 from the overnight M9-culture and incubated at 37 °C under shaking at 220 rpm. Antibiotics were added as required: kanamycin (50 μg ml$^{-1}$), ampicillin (100 μg ml$^{-1}$), and spectinomycin (100 μg ml$^{-1}$). For FACS (fluorescence-activated cell sorting) measurement, 10,000 cells were sorted per sample by BD LSRFortessa SORP flow-cytometer (BD Biosciences, USA). A 488-nm laser (blue) at 100 mW was used for green fluorescent with a 510/20 band pass filters. BD FACS Diva software (BD Biosciences, USA) and Flow Cytometry GUI for Matlab (version 1.3.0.0) by Nitai Steinberg (https://ww2.mathworks.cn/matlabcentral/fileexchange/38080-flow-cytometry-gui-for-matlab) were used for the analysis of acquired data.

**Glycerol measurements.** Glycerol was measured in the culture supernatant with a glycerol enzyme assay kit (MAK117-1KT, Sigma). 10 μl supernatant were mixed with 100 μl reaction buffer and incubated for 20 min. Absorbance was measured at 570 nm in a plate reader (Epoch, BioTek Instruments Inc, USA).

**Quantification of carotenoids.** Carotenoid production strains were cultivated in 96-well plates with M9 minimal medium containing 0.5% glucose and an additional 20% LB at 37 °C with continuous shaking. After 24 h cultivation, cells were harvested by centrifugation at 3220 × *g*. Cell pellets were resuspended in 120 μl DMSO[43], and sonicated for 30 s. Samples were centrifuged again at 3220 × *g* and 50 μl of the supernatants were transferred to a 384-well plate, and carotenoids were quantified by measuring the absorbance at 470 nm. For absolute quantification, standards of β-carotene (C4582-25MG, Sigma) were prepared at final concentrations of 5, 10, 25, and 50 mg l$^{-1}$.

**Metabolomics measurements.** Shake flask cultivations on M9 glucose were performed as described above. For steady-state metabolomics, cells were grown to an OD$_{600}$ of 0.5 and 2 ml culture aliquots were vacuum-filtered. For time-course metabolomics, volumes of samples were adjusted based on the OD$_{600}$ of the culture to obtain 1 ml with OD$_{600}$ 1. Culture aliquots were immediately filtered on a 0.45 μm pore size filter (HVLP02500, Merck Millipore) and filters were transferred into an extraction solution consisting of acetonitrile/methanol/water (40:40:20 (v/v)). Extracts were centrifuged for 20 min at −9 °C at 17,000 × *g* to remove the cell debris. Centrifuged extracts were mixed with $^{13}$C-labeled internal standard and analyzed by LC–MS/MS, with an Agilent 6495 triple quadrupole mass spectrometer (Agilent Technologies)[44]. An Agilent 1290 Infinity II UHPLC system (Agilent Technologies) was used for liquid chromatography and controlled by the Agilent MassHunter Acquisition software (Version B.07.01). The temperature of the column oven was 30 °C, and the injection volume was 3 μl. LC solvents A were water with 10 mM ammonium formate and 0.1% formic acid (v/v) (for acidic conditions); and water with 10 mM ammonium carbonate and 0.2% ammonium hydroxide (for basic conditions). LC solvents B were acetonitrile with 0.1% formic acid (v/v) for acidic conditions and acetonitrile without additive for basic conditions. LC columns were an Acquity BEH Amide (30 × 2.1 mm, 1.7 μm) for acidic conditions, and an iHILIC-

Fusion(P) (50 × 2.1 mm, 5 μm) for basic conditions. The gradient for basic and acidic conditions was: 0 min 90% B; 1.3 min 40% B; 1.5 min 40% B; 1.7 min 90% B; 2 min 90% B. Quantification of intracellular metabolite concentrations was based on the ratio of $^{12}$C and $^{13}$C peak heights.

**Proteomics measurements.** Cultivations were performed as described above. Culture aliquots were transferred into 2 ml reaction tubes and washed two times with PBS buffer (0.14 mM NaCl, 2.7 mM KCl, 1.5 mM KH$_2$PO$_4$, and 8 mM Na$_2$HPO$_4$). After washing, cell pellets were resuspended in 200 μl of lysis buffer containing 100 mM ammonium bicarbonate and 0.5 % sodium laroyl sarcosinate. Cells were again incubated for 15 min with 5 mM Tris(2-carboxyethyl)phosphine (TCEP) at 95 °C followed by alkylation with 10 mM iodoacetamide for 30 min at 25 °C. We used SP3 bead method[45,46] for a large number of samples. Fixed protein amount of 50 μg measured by BCA assay (23225, Thermo Fischer) and mixed with 4 μl SP3 beads stock (mixed 20 μl of each Sera-Mag Beads A and B (GE Healthcare) with 100 μl ddH$_2$O) in 96-well high-volume v-bottom plate (710879, Biozym Scientific GmbH). To initiate protein binding to the beads, 75 μl of 100% ethanol were added with the mixture of protein and beads for 15 min at room temperature. Tubes were placed in a magnetic rack for 5 min. The supernatant was discarded and the beads were rinsed two times with 200 μl of 70% ethanol and then 180 μl of 100% ethanol on a magnetic rack. For proteolytic digest, tubes were removed from the magnetic rack, and the beads were reconstituted in 28 μl 10% acetonitrile/10 mM NH$_4$HCO$_3$ with 1 μg trypsin (Promega) incubated shaking overnight at 30 °C. After incubation, the tubes were sonicated for 30 s and placed on a magnetic rack, and the supernatant containing tryptic peptides was recovered and transferred to new tubes. Recovered peptides were acidified by adding trifluoroacetic acid (TFA) to a 1.5% final concentration. Peptides were then purified through C18 microspin columns (Harvard Apparatus) according to the manufacturer's instruction. The eluted peptides were dried and resuspended in 0.1% TFA for analysis of peptides. Analysis of peptides was performed by a Q-Exactive Plus mass spectrometer connected to an Ultimate 3000 RSLC nano with a Prowflow upgrade and a nanospray flex ion source (Thermo Scientific) as previously described[32,47]. Briefly, peptides were separated by a reverse-phase HPLC column (75 μm × 42 cm) packed with 2.4 μm C18 resin (Dr. Maisch GmbH, Germany) at a flow rate of 300 nl/min by gradient model which is from 98% solvent A (0.15% formic acid) and 2% solvent B (99.85% acetonitrile, 0.15% formic acid) to 35% solvent B over 84 min. The data acquisition was set to obtain one high-resolution MS scan at a resolution of 70,000 full width at half maximum (at $m/z$ 200) followed by MS/MS scans of the 10 most intense ions. Label-free quantification (LFQ) of the data acquired from mass spectrometry was processed with Progenesis QIP (Waters), and MS/MS search was performed in MASCOT (v2.5, Matrix Science). The following search parameters were used: full tryptic search with two missed cleavage sites, 10 ppm MS1 and 0.02 Da fragment ion tolerance. Carbamidomethylation (C) as fixed, oxidation (M), and deamidation (N,Q) as variable modification. Progenesis outputs were further processed with SafeQuant.

**Constraint-based modeling.** Flux balance analysis (FBA) was performed with a genome-scale model of $E.\ coli$ metabolism, iML1515[19], and COBRApy[48]. Two additional reactions (G3PD_synth, G3PT_synth) were added to simulate the synthetic glycerol pathway. Constraints of the GLYK reaction were set to zero, to simulate deletion of the $glyK$ gene. Additionally, constraints of glycerol-3-phosphate and glycerol dehydrogenases G3PD5, G3PD6, G3PD7, and GLYCDx were set to zero. The model was further constrained to stimulate growth on a minimal medium, with glucose as the sole carbon source at an uptake rate of 8 mmol g$^{-1}$ h$^{-1}$. The oxygen uptake rate was constrained at maximum of 20 mmol g$^{-1}$ h$^{-1}$ and uptake of inorganic ions was not constrained (nh4, pi, so4, k, fe2, mg2, ca2, cl, mn2, zn2, ni2, cobalt2, mobd)[19]. FBA was performed with different glycerol production rates between 0 and 14 mmol g$^{-1}$ h$^{-1}$ and maximal growth was the objective function.

**Kinetic modeling and steady-state analysis.** The stoichiometry of the model is shown in Fig. 3a. Mass balancing yields a system of ordinary differential equations (ODEs), $F$, that is a temporal function of the state variables $x$ and the kinetic parameters $p$:

$$F(x,p) = \frac{dx}{dt} = \begin{cases} \frac{dFBP}{dt} = r_{\text{upper\_glycolysis}} - r_{\text{lower\_glycoylsis}} - r_{\text{glycerol}} - FBP \cdot \mu \\ \frac{de_1}{dt} = \beta_1 - e_1 \cdot \mu \\ \frac{de_2}{dt} = \beta_2 - e_2 \cdot \mu \end{cases} \quad (1)$$

The metabolite FBP is produced by $r_{\text{upper\_glycolysis}}$ and consumed by $r_{\text{lower\_glycolysis}}$ and $r_{\text{glycerol}}$. Additionally, FBP is diluted by growth. The enzyme e1 is a lower glycolysis enzyme for which we used parameters of glyceraldehyde-3-phosphate dehydrogenase (GapA) and e2 is GPD1. Both enzymes are produced by a production term $\beta$ and they are removed by dilution by growth. We assumed that enzyme degradation contributes little to the overall enzyme turnover and therefore can be neglected.

An upper glycolytic flux of 4.904 mmol g$^{-1}$ h$^{-1}$ was estimated with FBA using a glucose uptake rate of 8 mmol g$^{-1}$ h$^{-1}$. With the specific cell volume for $E.\ coli$

(2 μl mg$^{-1}$)[49] the reaction rate $r_{\text{upper\_glycolysis}}$ is:

$$r_{\text{upper\_glycolysis}} = \frac{4.904 \text{ mmol g}^{-1} \text{ h}^{-1}}{0.002 \text{ l g}^{-1}} * \frac{h}{60 \text{ min}} = 40.87 \text{ mM min}^{-1} \quad (2)$$

The reactions $r_{\text{lower\_glycolysis}}$ and $r_{\text{glycerol}}$ follow Michaelis–Menten kinetics:

$$r_{\text{lower\_glycolysis}} = k_{\text{cat,1}} \cdot e_1 \cdot \frac{FBP}{FBP + K_{m1}} \quad (3)$$

$$r_{\text{glycerol}} = k_{\text{cat,2}} \cdot e_2 \cdot \frac{FBP}{FBP + K_{m2}} \quad (4)$$

The expression rates of enzyme 1 (GapA) and enzyme 2 (GPD1) are:

$$\beta_1 = \beta_{1,\max} \cdot \left( \frac{FBP}{FBP_{SS}} \right)^{\alpha 1} \quad (5)$$

and

$$\beta_2 = \beta_{2,\max} \cdot ind \cdot \left( \frac{FBP}{FBP_{SS}} \right)^{\alpha 2} \quad (6)$$

Cra-regulation was simulated with a power-law term $\left( \frac{FBP}{FBP_{SS}} \right)^{\alpha}$ that affects the maximal enzyme expression rate. The power-law format has the advantage that the power-law term equals one in the un-induced state and therefore allows the same parameter values for the base model, the $\Delta cra$ model, and the 2× $cra$ model. Further, setting α to zero removes the regulation and therefore α2 was zero in the base model, while α1 and α2 were zero in the $\Delta cra$ model.

We assumed that the growth rate μ is proportional to $r_{\text{lower\_glycolysis}}$, because flux balance analysis showed a linear relationship between $r_{\text{lower\_glycolysis}}$ and the growth rates (Supplementary Fig. 11). Additionally, previous 13C labeling data showed a positive correlation between lower glycolytic flux and growth in $E.\ coli$[50]. With a growth rate of 0.01 min$^{-1}$ in the un-induced state, the proportionality factor-alpha follows as:

$$r_{\text{lower glycolysis}} = r_{\text{upper glycolysis}} - FBP \cdot \mu = 40.86 \text{ mM min}^{-1} \quad (7)$$

$$alpha = \frac{r_{\text{lower\_glycolysis}}}{\mu} = \frac{40.86 \text{ mM min}^{-1}}{0.01 \text{ min}^{-1}} = 4086 \text{ mM} \quad (8)$$

$$\mu = \frac{r_{\text{lower\_glycolysis}}}{alpha} \quad (9)$$

In total, the model includes 8 kinetic parameters $k_{\text{cat,1}}$, $k_{\text{cat,2}}$, $K_{m,1}$, $K_{m,2}$, $\beta_{1,\max}$, $\beta_{2,\max}$, α1, and α2. The parameters were either sampled 5000 times log-uniformly from predefined intervals or calculated based on steady state constraints. $K_{m,1}$ and $K_{m,2}$ were randomly sampled between 0.01 and 10 mM to account for high and low saturation of enzymes.

The power-law exponents α1 and α2 were randomly sampled between 1 and 2. The lower bound was 1 to ensure that the expression rate is at least linearly dependent on the FBP concentration. The upper bound was 2 to avoid higher-order dynamics that can cause instabilities[51].

The $k_{\text{cat,2}}$ value was based on the kinetic parameter of GPD1 ($k_{\text{cat,2}} = 1705$ min$^{-1}$)[52] and was sampled between 0.33-fold and 3-fold of this literature value. The parameter $k_{\text{cat,1}}$ followed from the steady-state constraint of the un-induced state where $r_{\text{glycerol}} = 0$.

$\beta_{1,\max}$ was derived from the mass balances of $e_1$, assuming steady state:

$$\beta_{1,\max} = e_1 \cdot \mu \quad (10)$$

The concentration of e1 was 0.0238 mM, based on quantitative proteome data for GapA[53] resulting in $\beta_{1,\max} = 0.000238$ mM min$^{-1}$.

The maximal enzyme expression rate in the glycerol pathway ($\beta_{2,\max}$) was defined by the translation rate of ribosomes according to:

$$\beta_{2,\max} = \frac{r_t \cdot R_{0.6} \cdot f_R}{L \cdot N_A \cdot V_c} \cdot p = 0.0017 \text{ mM min}^{-1} \quad (11)$$

Equation (11) considers the following parameters that were derived from the Bionumbers Database[54]: average translation rate ($r_t = 8.4$ amino acids s$^{-1}$), the median and abundance weighted protein length ($L = 209$ amino acids), the fraction of active ribosomes ($f_R = 0.8$) and the cellular volume ($V_{c,0.6} = 3 \times 10^{-15}$ L) and at a growth rate of $\mu = 0.6$ h$^{-1}$, the Avogadro number ($N_A = 6.02 \times 10^{23}$ mol$^{-1}$), the number of ribosomes per cell at that growth rate ($R_{0.6} = 8000$ ribosomes cell$^{-1}$). The fraction of ribosomes ($p$) that synthesize GPD1 at full induction was assumed to be 20%, because only 50% of the ribosomes can translate a heterologous protein and this is associated with significant protein burden[55].

**Steady state and robustness analysis.** To obtain steady states of the un-induced system, $\beta_{2,\max}$ and e2 were set to zero. Then 6 parameters were randomly sampled from intervals defined above. The 2 parameters ($k_{\text{cat1}}$ and $b_{1,\max}$) were calculated to ensure steady-state conditions. To test the stability of the steady states, eigenvalues of the Jacobian matrix were calculated, and tested if all eigenvalues are negative ($\lambda < -10^{-5}$). The procedure was repeated until 5000 stable steady states were achieved. Next, induction (ind in Eq. (6)) was iteratively increased from 0 to 1 using a numerical parameter continuation method. The method is based on finding a connected path of

steady-state concentrations ($x_{ss}$: steady-state concentration vector containing $e_{1ss}$, $e_{2ss}$, $FBP_{ss}$) as a parameter $p$ is varied. As the system is in steady state it follows that:

$$\frac{dx}{dt} = F(x_{ss}, p) = 0 \quad (12)$$

The derivative of $F(x_{ss}, p)$ with respect to the parameters is also zero:

$$\frac{dF(x_{ss}, p)}{dp} = \frac{\delta F}{\delta x_{ss}} \cdot \frac{dx_{ss}}{dp} + \frac{\delta F}{\delta p} = 0 \quad (13)$$

After rearranging Eq. (13), Eq. (14) is obtained:

$$\frac{dx_{ss}}{dp} = -\left(\frac{\delta F}{\delta x_{ss}}\right)^{-1} \cdot \frac{\delta F}{\delta p} \quad (14)$$

which describes the changes in the steady-state concentrations as a kinetic parameter that is varied iteratively. The iteration stops when one of the following two stability criteria is no longer fulfilled. 1st criterion: all real parts of the eigenvalues of the system's Jacobian need to be negative. In Eq. (14), the inverse of the Jacobian Matrix ($\delta F/\delta x_{SS}$) is required. The inversion is only possible as long as the matrix is regular. Once an eigenvalue reaches zero, the Jacobian becomes singular and matrix inversion is no longer possible. This bifurcation point defines the boundary between the stable and unstable parameter space. In other words: after this point is passed, the system cannot return to a stable steady state. Calculating the eigenvalues of the Jacobian at each step ensures that the iteration is terminated when one eigenvalue exceeds $\lambda = -10^{-5}$. The 2nd criterion is that all variables are positive.

**Quantification and statistical analysis**. Statistical analysis was performed with Matlab (R2018b) and custom Matlab scripts. The number of replicates ($n$) of each experiment can be found in the respective figure caption. For proteomics and metabolomics $n$ represents the number of independent shake flask cultures. In growth assays, $n$ represents the number of independent microtiter plate cultures.

**Reporting summary**. Further information on research design is available in the Nature Research Reporting Summary linked to this article.

## Data availability

Proteome raw data are deposited at the PRIDE database under the accession numbers: PXD022965. Metabolome raw data and flow cytometry data are deposited at the Open Research Data Repository of the Max Planck Society (Edmond) at https://edmond.mpdl.mpg.de/imeji/collection/67E9ibZDJRkXQh1s. Source data are provided with this paper.

## Code availability

The Matlab code for the model analysis is available at the GitHub repository (https://github.com/nfarke/Wang_Lempp_et_al).

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

## Acknowledgements
We thank T.J. Erb and L. Søgaard-Andersen for discussion. We thank B.S. Chen and V. Sourjik for providing plasmids and Dr. B. Ni for flow cytometry measurement. This research was supported by Deutsche Forschungsgemeinschaft grant LI 1993/2-1. H.L., C.-Y.W., and N.F. acknowledge funding from the Cluster of Excellence EXC 2124 from the Deutsche Forschungsgemeinschaft.

## Author contributions
C.-Y.W designed and performed experiments, analyzed data, and co-wrote the manuscript. M.L. analyzed data and performed model analysis and co-wrote the manuscript. N.F. performed model analysis. S.D performed FBA analysis. T.G. measured proteomes. H.L. coordinated the project, analyzed data, and wrote the manuscript.

## Funding

## Competing interests
The authors declare no competing interests.
