## [Peer Review File · Nature Communications]

Reviewers' Comments:

Reviewer #1:

Remarks to the Author:

The authors describe the use of metabolite feedback regulation to seamlessly integrate a heterologous pathway with native *E. coli* metabolism. By comparing static and dynamically regulated glycerol production pathways, ultrasensitivity was demonstrated at the interface between the synthetic pathway and glycolysis/gluconeogenesis.

By negatively regulating the synthetic pathway with a synthetic fructose-1,6-bisphosphate responsive pBAD promoter, expression of the synthetic glycerol production pathway was kept below levels that would cause *E. coli* metabolism to switch between glycolysis and gluconeogenesis by draining FBP. This alleviated the severe growth defect that was seen when the pathway was statically expressed, and enabled a higher glycerol titer.

As the authors state, dynamic sensor regulators have previously been engineered to control heterologous metabolism. The higher product titers that these approaches achieve have largely been attributed to the alleviation of resource burden from synthetic pathway expression and from avoiding product/intermediate toxicity. The novelty of this work therefore lies in the investigation of the metabolic causes and consequences of dynamic feedback regulation in a model synthetic pathway. In the case of glycerol production, regulation of glycolytic and gluconeogenic fluxes are fairly convincingly shown to underlie a growth defect from static pathway expression.

The use of FBP regulated promoters as a general tool for flux limited heterologous pathway expression is also useful and novel. Much of synthetic biology has been focused on controlling heterologous protein expression levels. In the field of metabolic engineering, there is a dawning realisation that rationally or semi-rationally designing these expression levels is inappropriate in the dynamic context of the cell. Due to the ubiquitous nature of glycolysis/gluconeogenesis and its conserved regulation via FBP, it is likely that many heterologous metabolite production pathways. These results on automatic dynamic control in the context of glycolytic flux regulation are therefore timely and important in a broad context.

The experiments are expertly conducted and portrayed in the manuscript, which is easy to follow despite some of the complex content. The conclusions are well supported by the data and correctly interpreted with appropriate statistical analyses. However, there are several points that should be addressed to improve the manuscript.

Major points:

1. Given that inappropriate gluconeogenic fluxes are likely the cause of the growth defect and lower glycerol titers, does deletion of *Cra* also solve these problems? It could even lead to higher performance if GPD1 and GPD2 can be expressed at higher levels using arabinose.
2. The main evidence that the synthetic pBAD/*Cra* promoter enables correct glycolytic fluxes is from FBP measurement (Figure 5 d). The manuscript would be greatly improved if more empirical evidence could be provided (the modelling is very nice, but it is modelling). Ideally this would be through the same metabolomics analyses as depicted in Figure 2 b, but I understand this is a large undertaking. It is quite interesting that FBP levels still decrease in the feedback responsive strain (Figure 5 d), and I wonder what the effects on the other glycolytic metabolite levels are.
3. Figure 1 b could be replicated with the synthetic pBAD/*Cra* promoter as part of Figure 4 to show that it leads to lower GPD1 expression levels. At the moment this is an obvious conclusion from the work, which is not directly shown or discussed. Alternatively, RT-qPCR on GPD1 mRNA could be used in the control and pBAD/*Cra* strain.
4. Does the synthetic *Cra* regulation system easily interface with other more industrially relevant *E. coli* promoters? I.e. can the operator sites be inserted into strong-constitutive promoters, or is it specific to the pBAD promoter. This is important for claiming broad utility of the system.

Minor points:

5. It would be useful for the mechanism behind FBP/Cra mediated pBAD promoter regulation to be explicitly discussed and illustrated with a figure. Presumably Cra binds to the promoter to block transcription when FBP is low, and binds to FBP but not the promoter when FBP is high, leading to transcription from pBAD in the presence of arabinose?

6. At the end of the introduction or start of the results section some rationale for industrial biological glycerol production could be provided (if it exists).

7. Page 3, line 76. The words 'glycolytic' and 'metabolite' are concatenated.

8. Page 7, line 188. The modelling of the glycerol pathway 'supports' rather than 'confirms' the hypothesis.

9. The sentence starting on page 8, line 198 needs rewording. The section stating "... showed almost a linear relationship that we expected in a first place" could be re-phrased as "... showed a near linear relationship, as we originally expected".

10. Page 8 line 205. No comma is required.

11. In Figure 5 b – d, the y-axes are labelled as fold change. Presumably this is relative to the non-induced control strain, but I couldn't see that mentioned anywhere?

12. Page 12 starting on line 346. It is mentioned that pure beta-carotene was used as a pure standard for absolute quantification. However, I did not see any mention of chromatographic carotenoid separation or any quantification other than absorbance in the manuscript.

Reviewer #2:

Remarks to the Author:

Feedback-regulated promoters enable robust control of synthetic metabolic pathways
Chun-Ying Wang, Martin Lempp, Niklas Farke, Stefano Donati, Timo Glatter, Hannes Link

Summary

The manuscript modifies a commonly used promoter (pBAD) to improve its control for use in synthetic circuits, and provides a nice proof of concept demonstration to indicate this might be more broadly applied. The study also provides some supporting experiments to demonstrate why such modifications are effective. Figures 2, 5 and 6, and the related results sections are fine, except for the comments below. There are some issues with Fig. 4 as noted below.

The main flaw of the manuscript is a misunderstanding of ultrasensitivity. There is no evidence of ultrasensitivity (including Fig 1), and the study did not properly measure the responses curves to make such a claim. Fig. 3 is confusing and I have some serious concerns about the model. Other claims in the abstract (see last comment below) provide a misleading impression of the study.

Comments

1. I strongly recommend the authors remove all claims of ultrasensitivity from the paper, and simply state what is observed - growth inhibition at higher levels of induction.

A prominent claim of the paper is that the system displays ultrasensitivity. Ultrasensitivity generally means that in a biochemical system the increase in the concentration of Y (output) in response to the change in concentration X (input) is much greater than expected with a positive cooperative system (see references below). For example, in Koshland's original papers an ultrasensitive system generates a response curve that is the equivalent of a Hill coefficient of 81 (for comparison the Hill coefficient for hemoglobin is about 3).

The manuscript states that Fig. 1D demonstrates ultrasensitivity, but I do not see where the output of the gene (concentration Y) increases steeply as a function of arabinose (concentration

X). It could be much clearer in the paper what evidence is used to make this claim. Fig. 1D has three panels. I assume the claim is not based on the optical density increasing as a function of time (left panel) or the decrease in growth rate with arabinose induction (center panel), but on the peak in glycerol concentration within a range of the arabinose concentration (right panel), which is due to the toxic effect of higher levels of induction of *gpd1* and *gpp2*. I agree that the concentration of arabinose that gives maximal induction for these genes is at a lower concentration than the *gfp* control, but this is because higher levels of expression for these genes are not permissible rather than the actual induction curve being steeper. On the last point, there is evidence that without the toxic effects (indicated by the decrease in cell growth) that dose response curve would have a similar slope to the control - see right column of plots in Fig. 4 at about 0.3% arabinose (the concentration of glycerol in all plots is similar). Therefore, the authors have not actually shown that the system generates ultrasensitivity. They have only shown that it is responsive to low(ish) percentage of arabinose. As an aside, 0.2% arabinose = 2g of arabinose in 1 L = 13 mM, which is actually quite a lot of a chemical inducer (arabinose has Mwt of ~150 g/mol).

Further, on the point of ultrasensitivity, the authors would need additional measurements with more resolution in the range 0-0.2% arabinose and fit the data to a Hill function or measure the logarithmic gain to demonstrate ultrasensitivity. Interpretation of the response curve is also challenging because (1) they are measuring by OD (which may not be a linear measure of cell number especially at the high values in Fig. 1D) and (2) because of the effect of growth rate which affects exponential decay/dilution of the proteins and molecules. To elaborate on the last point, the measurement of glycerol at 0.2% and 2% arabinose in the right panel of Fig. 1D not only depends on their level of induction but also their dilution (which is also changing with the induction).

It is unclear that the effect of cell number was taken into account for the glycerol measurement in the supernatant - there was no mention of a correction for cell number or OD on page 13.

References:

Zhang Qiang, Bhattacharya Sudin and Andersen Melvin E. 2013 Ultrasensitive response motifs: basic amplifiers in molecular signalling networks. *Open Biol.* 3:130031
Ferrell JE Jr, Ha SH. Ultrasensitivity part I: Michaelian responses and zero-order ultrasensitivity. *Trends Biochem Sci.* 2014;39(10):496-503.

2. The model and Figure 3 do not provide a convincing explanation for ultrasensitivity and there are concerns with both. The results section states there is ultrasensitivity but there is no evidence provided. By what metric or criteria is a response curve being determined as "ultrasensitive"? Figure 3 shows the ratio of the two competing output pathways as a function of B2 induction, which is not an obvious metric to show or explain the ultrasensitivity. Readers shouldn't be looking at the curves and trying to judge whether they are ultrasensitive. Please see the references above for metrics.

Because the steepest part of the response is occurring at low concentrations, I would recommend the authors plot the data in Figure 3b on a log x-axis. I am concerned that a shift in the model or data that decreases the dissociation constant will appear to make the curve steeper but it hasn't. For example, if the experiments or simulations for the arabinose concentrations are in step sizes of 0.1%, then decreasing the K_d from 20 to 5 could make it appear that there is a steeper function on a linear x-axis plot even if the Hill coefficient or logarithmic gain is unchanged.

It is very unclear what Fig 3a is intended to show. Is it to make the case that there is competition for a finite resource (m), and therefore in a manner similar to futile cycles there is generation of ultrasensitivity? Or that with regard to the ratio of the enzymes, both increasing e_2 and decreasing e_1 , results in a larger change in the ratio of the product of the two pathways? There needs to be a more convincing explanation for any ultrasensitive response.

There are some concerns about the model. First, the model needs more explanation - so, I provide the following comments recognizing that I may be missing something important. Equations 3 and 4 should be dependent on the concentration of m (which is a metabolite according to the Fig. 3

legend), otherwise dm/dt in Equation 1 can result in negative concentrations of m . This could be a cause of the model's instability mentioned in the figure legend. I think there might also be an issue with the units for Eq. 3 and 4. e has units of mM of enzyme (presumably, otherwise Eq 11 does not make sense), therefore there are no units of mM for metabolite m in Eq 3 and 4. Note: while k_{cat} usually has units of min^{-1} this usually has an implied unit of per enzyme. In this case, a more appropriate unit would for k_{cat} would be min^{-1} per mM of enzyme, so that $r = k_{cat} * [e] * [m] * m / (m + k_m)$ has units of mM (of m) min^{-1} or to have e in units of number of molecules.

I recommend removing Fig. 3 and this model.

3. Figure 4 is good figure, especially 4C, which eliminated many of my concerns that 4A and 4B might be a non-specific effect of the sequence changes. But I do not really understand why selective inhibition of *gpd1* and not *gpp2* via the Cra binding site was chosen. Would putting the Cra site in the *gpp2* promoter have the same effect? An additional useful control experiment would be to mutate the pBAD promoter to decrease transcription without adding the Cra binding site. Does mutating the -10 and -35 sites have the same effect? This is a crucial point – is it feedback or simply lower maximum transcription from pBAD-Cra that is responsible for the reduction in toxicity?

4. Figure 6 is a good demonstration of the system. These points need to be addressed: (1) Fig 6b x-axis label is needed, (2) please indicate in the Fig. 6 legend what the grey shaded area indicates in panel b, (3) Are the carotenoid measurements in Fig. 6 normalized by OD? If not, they should be, otherwise it is not clear to what extent the concentration of carotenoid drops because the cells are sick.

5. A figure showing the regulatory circuit with FBP, Cra and the other genes mentioned (eg PpsA) would help the reader to better understand the feedback regulation.

6. The main point seems to be that feedback is useful to prevent the accumulation of toxic intermediates or depletion of necessary metabolites in expressed systems. If this correct, this point could be made much clearer otherwise the reader may have a hard time connecting the results to the title and abstract. The short discussion did not help much in this regard. Also, the experiment with the weaker promoter suggested in Point 3 would help address this point.

7. These sentences in the Abstract should be re-written. "With static promoters, *E. coli* glycolysis is ultrasensitive against expression rates of enzymes in the synthetic glycerol pathway. Ultrasensitivity emerges from a regulatory interaction between fructose-1,6-bisphosphate (FBP) and the transcription factor Cra, which switches between glycolysis and gluconeogenesis in *E. coli*. This regulation effectively prevents to establish an optimal trade-off between glycolysis and the synthetic glycerol pathway." The study does not show ultrasensitivity, and it has not properly shown that there is a trade-off. I agree that there could be a trade-off, which would explain the results, but the author have actually not demonstrated that and therefore should not be claiming it in the abstract. To demonstrate a trade-off, experiments could be performed to show a negative correlation between the output of the glycolysis and gluconeogenesis pathways, or a phase diagram or heat map with glycolysis and gluconeogenesis on the axes and growth rates in that space.

We thank the reviewers for their time to review our manuscript and their constructive comments, on which we based new experiments and revisions of the manuscript. A point by point response is below and the most important changes are:

- We removed all claims and conclusions about ultrasensitivity. Instead, we focus on the observation that glycerol production causes a growth burden because metabolite levels decreased, and that feedback regulation prevents this. We highlight this in a new title: *“Feedback-regulated promoters enable robust control of synthetic metabolic pathways by stabilizing metabolite levels”*
- We expressed the glycerol pathway with a weaker pBAD promoter, which had mutations in the -10 and -35 region. The results show that changing the basal promoter activity has smaller effects on robustness than feedback regulation by Cra (**new Figure 4a**).
- We confirmed the key assumption that the pBAD-Cra promoter leads to lower GPD1 expression levels. Therefore we measured activity of the pBAD-Cra promoter by expressing a GPD1-GFP fusion protein (**new Supplementary Figure 4**).
- We expressed the glycerol pathway in a *cra* deletion strain (Δ *cra* strain). The Δ *cra* strain is indeed better than the base strain, which further strengthens our main hypothesis that Cra-regulation causes a growth burden in the production strain (**new Figure 2e**).
- We measured the metabolome of the Cra-regulated strain under steady state conditions. The data supports our assumption that FBP levels (and other glycolysis metabolites) remain higher in the pBAD-Cra strain than in the base strain (**new Supplementary Figure 3**).
- To demonstrate broad utility of the system, we inserted Cra into a constitutive promoter and an aTc inducible pTet promoter (**new Supplementary Figure 6**).
- We restructured the model analysis and used a robustness analysis to have a clear metric about performance of the different models (**new Figure 3**).

Response to Reviewer #1

The authors describe the use of metabolite feedback regulation to seamlessly integrate a heterologous pathway with native *E. coli* metabolism. By comparing static and dynamically regulated glycerol production pathways, ultrasensitivity was demonstrated at the interface between the synthetic pathway and glycolysis/gluconeogenesis.

By negatively regulating the synthetic pathway with a synthetic fructose-1,6-bisphosphate responsive pBAD promoter, expression of the synthetic glycerol production pathway was kept below levels that would cause *E. coli* metabolism to switch between glycolysis and gluconeogenesis by draining FBP. This alleviated the severe growth defect that was seen when the pathway was statically expressed, and enabled a higher glycerol titer.

As the authors state, dynamic sensor regulators have previously been engineered to control heterologous metabolism. The higher product titers that these approaches achieve have largely been attributed to the alleviation of resource burden from synthetic pathway expression and from avoiding product/intermediate toxicity. The novelty of this work therefore lies in the investigation of the metabolic causes and consequences of dynamic feedback regulation in a model synthetic pathway. In the case of glycerol production, regulation of glycolytic and gluconeogenic fluxes are fairly convincingly shown to underlie a growth defect from static pathway expression.

The use of FBP regulated promoters as a general tool for flux limited heterologous pathway expression is also useful and novel. Much of synthetic biology has been focused on controlling heterologous protein expression levels. In the field of metabolic engineering, there is a dawning realisation that rationally or semi-rationally designing these expression levels is inappropriate in the dynamic context of the cell. Due to the ubiquitous nature of glycolysis/gluconeogenesis and its conserved regulation via FBP, it is likely that many heterologous metabolite production pathways. These results on automatic dynamic control in the context of glycolytic flux regulation are therefore timely and important in a broad context.

The experiments are expertly conducted and portrayed in the manuscript, which is easy to follow despite some of the complex content. The conclusions are well supported by the data and correctly interpreted with appropriate statistical analyses. However, there are several points that should be addressed to improve the manuscript.

We thank the reviewer for the positive comments and the constructive suggestions.

Major points:

1. Given that inappropriate gluconeogenic fluxes are likely the cause of the growth defect and lower glycerol titers, does deletion of *Cra* also solve these problems? It could even lead to higher performance if GPD1 and GPD2 can be expressed at higher levels using arabinose.

We agree with the reviewer that if Cra causes the growth defect, it should be solved by removing Cra. Therefore, we performed additional experiments and expressed the glycerol pathway in a *cra* deletion strain (Δ *cra*). The Δ *cra* strain was indeed more robust than the base strain. This observation further strengthened our claim that Cra regulation is responsible for the growth burden of glycerol production. We show these results in a **new Figure 2e**, and discuss it in the text:

*“To test this hypothesis we deleted *cra* and expressed the glycerol pathway in the resulting Δ *cra* strain (Fig. 2e). Compared to the base strain, growth of the Δ *cra* strain was more robust against induction of the synthetic glycerol pathway, and the maximum glycerol titers increased 1.6-fold (compare Fig. 2e and Fig. 1d). Thus, our data indicate a regulatory problem in glycerol overproducing *E. coli*: glycerol production perturbs FBP levels, which activate Cra and eventually gluconeogenesis.”*

2. The main evidence that the synthetic pBAD/Cra promoter enables correct glycolytic fluxes is from FBP measurement (Figure 5 d). The manuscript would be greatly improved if more empirical evidence could be provided (the modelling is very nice, but it is modelling). Ideally this would be through the same metabolomics analyses as depicted in Figure 2 b, but I understand this is a large undertaking. It is quite interesting that FBP levels still decrease in the feedback responsive strain (Figure 5 d), and I wonder what the effects on the other glycolytic metabolite levels are.

We performed the suggested experiment and measured the metabolome of the pBAD-Cra strain under steady state conditions at three induction levels. The data confirms that FBP levels in the pBAD-Cra strain remain higher than in the base strain. Also other glycolytic metabolites remain at higher concentrations in the pBAD-Cra strain. We show these results in a **new Supplementary Figure 3** and discuss it in the text:

*“Additionally, we confirmed that the pBAD-Cra strain maintained higher concentrations of FBP and other glycolytic metabolites under steady state conditions with constant induction at 0%, 0.1% and 0.5% *ara* (Supplementary Figure 3).”*

3. Figure 1 b could be replicated with the synthetic pBAD/Cra promoter as part of Figure 4 to show that it leads to lower GPD1 expression levels. At the moment this is an obvious conclusion from the work, which is not directly shown or discussed. Alternatively, RT-qPCR on GPD1 mRNA could be used in the control and pBAD/Cra strain.

We agree that the pBAD/Cra promoter must lead to lower expression levels of GPD1. The proteome data in Figure 5 included GPD1 and shows that GPD1 expression is lower in the

pBAD-Cra (**new Figure 5c**). Moreover, as suggested by the reviewer, we replicated Figure 1b and measured expression of a GFP-GPD1 fusion protein (we used the fusion protein instead of GFP to account for the additional effect of glycerol production). The results show that the pBAD/Cra promoter has indeed a lower activity than the pBAD promoter (**new Supplementary Figure 4**).

Additionally, the GPD1-GFP enabled us to show this effect at the single cell level and to confirm that all cells have lower expression. We show the flow cytometry results in a **new Supplementary Figure 5**.

“To compare activity of the pBAD promoter and the pBAD-Cra promoter in the context of our production system, we replaced GPD1 with a GPD1-GFP fusion protein (Supplementary Figure 4). The pBAD-Cra promoter had a 3.7-fold lower promoter activity than the pBAD promoter at 0.5% ara, showing that inserting the Cra binding site reduced GPD1 expression. Flow cytometry data revealed that the cell-to-cell variation in GPD1-GFP content was similar with the pBAD and the pBAD-Cra promoter (Supplementary Figure 5), which means that all cells of the pBAD-Cra population have a lower promoter activity.”

4. Does the synthetic Cra regulation system easily interface with other more industrially relevant E. coli promoters? I.e. can the operator sites be inserted into strong-constitutive promoters, or is it specific to the pBAD promoter. This is important for claiming broad utility of the system.

We agree that a broad utility is an important point and to test this we inserted the Cra binding site into: 1) a **constitutive promoter** and 2) the aTc inducible **pTet promoter**. We then expressed the glycerol pathway with the constitutive and constitutive/Cra promoter and the pTet and pTet/Cra promoter. In both cases, the promoter with Cra-binding site improved both growth and glycerol production, demonstrating the broad utility of our approach. We show these results in a **new Supplementary Figure 6**.

“To demonstrate the broad utility of this approach we inserted Cra into a constitutive promoter and the pTet promoter (Supplementary Figure 6). In both cases, the strain with the Cra-regulated promoter variant was more robust against induction of the glycerol pathway.”

Minor points:

5. It would be useful for the mechanism behind FBP/Cra mediated pBAD promoter regulation to be explicitly discussed and illustrated with a figure. Presumably Cra binds to the promoter to block transcription when FBP is low, and binds to FBP but not the promoter when FBP is high, leading to transcription from pBAD in the presence of arabinose?

We added an illustration of the regulation relationship between FBP and Cra in a **new Figure 2c**.

6. At the end of the introduction or start of the results section some rationale for industrial biological glycerol production could be provided (if it exists).

“According to the United States Department of Energy (DOE), glycerol belongs to top 10 value added chemicals from biomass¹⁶, and can be a precursor for other bio-based products like acrylic acid¹⁷ and 1,3-propanediol¹⁸”

7. Page 3, line 76. The words ‘glycolytic’ and ‘metabolite’ are concatenated.

Corrected: from the “glycolytic metabolite” dihydroxyacetone phosphate (DHAP).

8. Page 7, line 188. The modelling of the glycerol pathway ‘supports’ rather than ‘confirms’ the hypothesis.

Corrected: *“Thus a simple model supports our hypothesis that the transcriptional response by Cra is problematic in glycerol producing E. coli.”*

9. The sentence starting on page 8, line 198 needs rewording. The section stating “... showed almost a linear relationship that we expected in a first place” could be re-phrased as “... showed a near linear relationship, as we originally expected”.

We removed the sentence and rewrote the paragraph.

“Notably, by adding Cra-regulation we achieved the near linear relationship between growth rate and ara levels, as we originally expected (Fig. 3c).”

10. Page 8 line 205. No comma is required.

Corrected

11. In Figure 5 b – d, the y-axes are labelled as fold change. Presumably this is relative to the non-induced control strain, but I couldn’t see that mentioned anywhere?

Yes, it is relative to the 0h time point (un-induced) of the base strain. We added a description in the figure caption: *“Data is normalized to the 0 h time point of the base strain.”*

12. Page 12 starting on line 346. It is mentioned that pure beta-carotene was used as a pure standard for absolute quantification. However, I did not see any mention of chromatographic carotenoid separation or any quantification other than absorbance in the manuscript.

We did not separate carotenoids by a column but measured the total pool of carotenoids (in our case we expect a mixture of β -carotene, zeaxanthin and lycopene). We used carotene as a standard to estimate the absolute concentration of the carotenoid mixture. We only used carotene for calibration because the colored carotenoids (β -carotene, zeaxanthin and lycopene) show very similar absorption spectra⁴.

Response to Reviewer #2

Feedback-regulated promoters enable robust control of synthetic metabolic pathways
Chun-Ying Wang, Martin Lempp, Niklas Farke, Stefano Donati, Timo Glatter, Hannes Link

Summary

The manuscript modifies a commonly used promoter (pBAD) to improve its control for use in synthetic circuits, and provides a nice proof of concept demonstration to indicate this might be more broadly applied. The study also provides some supporting experiments to demonstrate why such modifications are effective. Figures 2, 5 and 6, and the related results sections are fine, except for the comments below. There are some issues with Fig. 4 as noted below.

The main flaw of the manuscript is a misunderstanding of ultrasensitivity. There is no evidence of ultrasensitivity (including Fig 1), and the study did not properly measure the responses curves to make such a claim. Fig. 3 is confusing and I have some serious concerns about the model. Other claims in the abstract (see last comment below) provide a misleading impression of the study.

We thank the reviewer for the constructive critique and the suggestions for additional experiments. They all helped us to focus and strengthen the main point of the manuscript. The claims about ultrasensitivity were indeed distracting and ambiguous, and we removed them throughout the revised manuscript.

Comments

1. I strongly recommend the authors remove all claims of ultrasensitivity from the paper, and simply state what is observed - **growth inhibition at higher levels of induction**.

A prominent claim of the paper is that the system displays ultrasensitivity. Ultrasensitivity generally means that in a biochemical system the increase in the concentration of Y (output) in response to the change in concentration X (input) is much greater than expected with a positive cooperative system (see references below). For example, in Koshland's original papers an ultrasensitive system generates a response curve that is the equivalent of a Hill coefficient of 81 (for comparison the Hill coefficient for hemoglobin is about 3).

The manuscript states that Fig. 1D demonstrates ultrasensitivity, but I do not see where the output of the gene (concentration Y) increases steeply as a function of arabinose (concentration X). It could be much clearer in the paper what evidence is used to make this claim. Fig. 1D has three panels. I assume the claim is not based on the optical density increasing as a function of time (left panel) or the decrease in growth rate with arabinose induction (center panel), but on the peak in glycerol concentration within a range of the arabinose concentration (right panel), which is due to the toxic effect of higher levels of induction of *gpd1* and *gpp2*.

I agree that the concentration of arabinose that gives maximal induction for these genes is at a

lower concentration than the gfp control, but this is because higher levels of expression for these genes are not permissible rather than the actual induction curve being steeper. On the last point, there is evidence that without the toxic effects (indicated by the decrease in cell growth) that dose response curve would have a similar slope to the control - see right column of plots in Fig. 4 at about 0.3% arabinose (the concentration of glycerol in all plots is similar). Therefore, the authors have not actually shown that the system generates ultrasensitivity. They have only shown that it is responsive to low(ish) percentage of arabinose. As an aside, 0.2% arabinose = 2g of arabinose in 1 L = 13 mM, which is actually quite a lot of a chemical inducer (arabinose has Mwt of ~150 g/mol). Further, on the point of ultrasensitivity, the authors would need additional measurements with more resolution in the range 0-0.2% arabinose and fit the data to a Hill function or measure the logarithmic gain to demonstrate ultrasensitivity.

Interpretation of the response curve is also challenging because (1) they are measuring by OD (which may not be a linear measure of cell number especially at the high values in Fig. 1D) and (2) because of the effect of growth rate which affects exponential decay/dilution of the proteins and molecules. To elaborate on the last point, the measurement of glycerol at 0.2% and 2% arabinose in the right panel of Fig. 1D not only depends on their level of induction but also their dilution (which is also changing with the induction).

It is unclear that the effect of cell number was taken into account for the glycerol measurement in the supernatant – there was no mention of a correction for cell number or OD on page 13.

References:

Zhang Qiang, Bhattacharya Sudin and Andersen Melvin E. 2013 Ultrasensitive response motifs: basic amplifiers in molecular signalling networks. *Open Biol.* 3:130031

Ferrell JE Jr, Ha SH. Ultrasensitivity part I: Michaelian responses and zero-order ultrasensitivity. *Trends Biochem Sci.* 2014;39(10):496-503.

Response to point 1)

We agree with the reviewer that ultrasensitivity is ambiguous and misleading in our context. We removed all claims about ultrasensitivity and instead focus on our observations: i) growth is inhibited at higher inducer levels, ii) this leads to low biomass levels and iii) the consequence are low glycerol titers. We think that this better describes the conclusion from data in Fig. 1d and changed the text, for example in the abstract:

“Expressing the synthetic glycerol pathway with static promoters caused a strong growth burden, resulting in low biomass-levels and low glycerol titers.”

Or the results around Fig 1d:

“However, low ara levels caused a growth burden in the base strain (Fig. 1d). Already 0.3% ara caused a strong defect and low biomass concentrations and titers of glycerol. The maximal glycerol titers were achieved with 0.1% ara (17.71 mM, Fig. 1d).”

We politely disagree with the reviewers concern about high levels of arabinose. First, other studies with the pBAD promoter used similar concentration ranges of arabinose (e.g. 0% to 1% arabinose in Ceroni et al. 2018, Nature Methods⁵). Second, we show that activity of the pBAD promoter changes linearly between the concentration range of 0% and 2% arabinose, and we show that growth is not affected by these arabinose levels (**Supplementary Figure 2**).

Regarding the concern about cell size effects: The metabolome and proteome data were normalized by OD, and we agree with the reviewer's point that this requires a constant relationship between cell number and OD. We used the side and forward scatter of our FACS data (at 0% 0.1% and 0.5% arabinose), to exclude tangible effects of glycerol production on cell size or heterogeneity of the culture. This result is shown in a **new Supplementary Figure 5**.

We did not normalize the glycerol titers by OD. We believe the main point of our work is to maintain high growth rates at high glycerol production rates. Only the combination of high biomass and high production leads to high titers, which is arguably the most important parameter in an industrial setup. Moreover, in Fig. 5b we show that specific glycerol production rates are higher in the pBAD-Cra strain.

2. The model and Figure 3 do not provide a convincing explanation for ultrasensitivity and there are concerns with both. The results section states there is ultrasensitivity but there is no evidence provided. By what metric or criteria is a response curve being determined as "ultrasensitive"? Figure 3 shows the ratio of the two competing output pathways as a function of B2 induction, which is not an obvious metric to show or explain the ultrasensitivity. Readers shouldn't be looking at the curves and trying to judge whether they are ultrasensitive. Please see the references above for metrics.

Because the steepest part of the response is occurring at low concentrations, I would recommend the authors plot the data in Figure 3b on a log x-axis. I am concerned that a shift in the model or data that decreases the dissociation constant will appear to make the curve steeper but it hasn't. For example, if the experiments or simulations for the arabinose concentrations are in step sizes of 0.1%, then decreasing the Kd from 20 to 5 could make it appear that there is a steeper function on a linear x-axis plot even if the Hill coefficient or logarithmic gain is unchanged.

It is very unclear what Fig 3a is intended to show. Is it to make the case that there is competition for a finite resource (m), and therefore in a manner similar to futile cycles there is generation of ultrasensitivity? Or that with regard to the ratio of the enzymes, both increasing e_2 and decreasing e_1 , results in a larger change in the ratio of the product of the two pathways? There needs to be a more convincing explanation for any ultrasensitive response.

There are some concerns about the model. First, the model needs more explanation – so, I provide the following comments recognizing that I may be missing something important. Equations 3 and 4 should be dependent on the concentration of m (which is a metabolite according to the Fig. 3

legend), otherwise dm/dt in Equation 1 can result in negative concentrations of m . This could be a cause of the model's instability mentioned in the figure legend. I think there might also be an issue with the units for Eq. 3 and 4. e has units of mM of enzyme (presumably, otherwise Eq 11 does not make sense), therefore there are no units of mM for metabolite m in Eq 3 and 4. Note: while k_{cat} usually has units of min^{-1} this usually has an implied unit of per enzyme. In this case, a more appropriate unit would for k_{cat} would be min^{-1} per mM of enzyme, so that $r = k_{cat} * [e] * [m] / (m + k_m)$ has units of mM (of m) min^{-1} or to have e in units of number of molecules.

I recommend removing Fig. 3 and this model.

Response to point 2)

We agree with the reviewers' point that we did not provide a sufficiently clear metric for performance of the different models. To solve this problem we completely restructured the modelling part and used an established robustness analysis method (EMRA) that was developed by the James Liao group. With this method we could quantify robustness of each model at each inducer level. The result shows that robustness of the base strain decreases with higher induction of the glycerol pathway, while the Δcra model and the model with additional Cra regulation remain robust. We replaced the former Fig. 3b with this robustness analysis.

We respectfully disagree with reviewers concern about the model:

- **Negative concentrations.** Equations 3 and 4 were correct and negative concentrations cannot occur. Both Eq 3 and 4 are simple Michaelis Menten kinetics, and they do depend on the concentration of the metabolite [FBP] and the enzyme. For clarity we also labelled the metabolite as FBP. As the reviewer mentions no dependency on FBP would lead to negative concentrations and instabilities. However this would affect all models, and not only the base strain model. We better describe this in the revised text: *"FBP influences reaction rates in lower glycolysis ($r_{\text{lower_glycolysis}}$) and in the glycerol pathway (r_{glycerol}) according to simple Michaelis Menten kinetics."*
- The units of Equations 3 and 4 were correct and all rates are given in mM min^{-1} . We apologize if units were not clear and to avoid confusions we added a **new Table 1** that summarizes units and intervals of parameters. For clarity we also used a power-law term which allowed us to use the same 1000 parameter sets for all models (base strain, Δcra , and 2x Cra model)

We considered the recommendation of removing the model. But we kept it because building predictive dynamic models is one of the main goals in the field. We believe that our model is a successful example, because it qualitatively captures our experimental observations: the base strain is not robust, while the Δcra strain and the 2x cra strain are more robust. To highlight this we

added the experimental data to Figure 3 and oppose it with the model results.

3. Figure 4 is good figure, especially 4C, which eliminated many of my concerns that 4A and 4B might be a non-specific effect of the sequence changes. But I do not really understand why selective inhibition of *gpd1* and not *gpp2* via the Cra binding site was chosen. Would putting the Cra site in the *gpp2* promoter have the same effect?

An additional useful control experiment would be to mutate the pBAD promoter to decrease transcription without adding the Cra binding site. Does mutating the -10 and -35 sites have the same effect? This is a crucial point – is it feedback or simply lower maximum transcription from pBAD-Cra that is responsible for the reduction in toxicity?

Response to point 3)

We control expression of GPD1 because it is the first enzyme in the pathway. Controlling the abundance of the second enzyme GPP2 would probably lead to accumulation of the intermediate glycerol-P, which could be toxic for example due to phosphate stress. Thus, we did not consider controlling the glycerol pathway in the middle. We illustrated this logic in Fig. 1c.

The reviewer points to a valid concern: a lower basal activity of the pBAD-Cra promoter could be responsible for the improvement of the glycerol producer. Although we believe that titration with *ara* accounts for such effects we followed the reviewers' suggestion and mutated the -10 and -35 region of the pBAD promoter:

Therefore we introduced mutations between the -10 and -35 region of the pBAD promoter, which were described in the literature and reduce its activity. Then we expressed the glycerol pathway with the original pBAD promoter and the weaker mutated pBAD promoter. The weaker pBAD promoter improved growth and glycerol titers, but the improvement was small compared the pBAD-Cra promoter. Thus, the weaker maximum activity of the pBAD-Cra promoter has a smaller effects on growth and glycerol production than its regulation by Cra. We show the results in Figure 4b and included it in the text:

“To exclude that the lower activity of the pBAD-Cra promoter increased robustness, we used a pBAD promoter with mutations between the -10 and -35 boxes that decreased activity by a factor of two²⁴. A glycerol producer with this weaker, mutated pBAD promoter was less robust than the pBAD-Cra strain (Fig. 4b), thus indicating that feedback regulation by Cra increased robustness and not lower promoter activity per se”

4. Figure 6 is a good demonstration of the system. These points need to be addressed: (1) Fig 6b x-axis label is needed, (2) please indicate in the Fig. 6 legend what the grey shaded area indicates in panel b, (3) Are the carotenoid measurements in Fig. 6 normalized by OD? If not, they should be, otherwise it is not clear to what extent the concentration of carotenoid drops because the cells are

sick.

Response to point 4)

(1) We added the x-axis label.

(2) We added the information to the figure caption: *“Growth curves and dots show means of n = 2 cultures. Grey shaded areas show the difference between two cultures”*

(3) The carotenoid titer and was not normalized by OD for the same reasons as mentioned above for glycerol titers.

5. A figure showing the regulatory circuit with FBP, Cra and the other genes mentioned (eg PpsA) would help the reader to better understand the feedback regulation.

Response to point 5)

We added a new Fig. 2c that illustrates the regulatory logic of Cra and FBP.

6. The main point seems to be that feedback is useful to prevent the accumulation of toxic intermediates or depletion of necessary metabolites in expressed systems. If this correct, this point could be made much clearer otherwise the reader may have a hard time connecting the results to the title and abstract. The short discussion did not help much in this regard. Also, the experiment with the weaker promoter suggested in Point 3 would help address this point.

Response to point 6)

We agree that the main point is preventing depletion of metabolites (especially regulatory metabolites like FBP). We made additional metabolite measurements in the pBAD-Cra strain to support this claim (Supplementary Figure 3). We also revised the text to make this point clearer, for example in the title:

*“Feedback-regulated promoters enable robust control of synthetic metabolic pathways **by stabilizing metabolite levels**”*

Or in the discussion:

“Here, we observed strong decreases of glycolytic metabolites in glycerol producing E. coli, especially of the regulatory metabolite FBP. Low FBP levels might be the reason for increases of gluconeogenic enzymes like PpsA, because low FBP levels can active the transcription factor Cra. This result highlights the importance of maintaining regulatory metabolites above a critical threshold in engineered microbes.”

7. These sentences in the Abstract should be re-written. “With static promoters, E. coli glycolysis is ultrasensitive against expression rates of enzymes in the synthetic glycerol pathway. Ultrasensitivity emerges from a regulatory interaction between fructose-1,6-bisphosphate (FBP) and

the transcription factor Cra, which switches between glycolysis and gluconeogenesis in *E. coli*. This regulation effectively prevents to establish an optimal trade-off between glycolysis and the synthetic glycerol pathway.” The study does not show ultrasensitivity, and it has not properly shown that there is a trade-off. I agree that there could be a trade-off, which would explain the results, but the author have actually not demonstrated that and therefore should not be claiming it in the abstract. To demonstrate a trade-off, experiments could be performed to show a negative correlation between the output of the glycolysis and gluconeogenesis pathways, or a phase diagram or heat map with glycolysis and gluconeogenesis on the axes and growth rates in that space.

Response to point 7)

We agree with the reviewer and changed the abstract accordingly (no claims about ultrasensitivity or a trade-off):

“Expressing the synthetic glycerol pathway with static promoters caused a strong growth burden, resulting in low biomass-levels and low glycerol titers. We show that the growth burden is due to a regulatory interaction between fructose-1,6-bisphosphate (FBP) and the transcription factor Cra, which switches between glycolysis and gluconeogenesis in E. coli. This regulation activated gluconeogenesis at higher induction of the glycerol pathway, because FBP levels were low.”

References

1. Werpy, T. & Petersen, G. Top value added chemicals from biomass: volume I--results of screening for potential candidates from sugars and synthesis gas. (2004).
2. Sun, D., Yamada, Y., Sato, S. & Ueda, W. Glycerol as a potential renewable raw material for acrylic acid production. *Green Chem.* **19**, 3186–3213 (2017).
3. Tabah, B. et al. Production of 1, 3-propanediol from glycerol via fermentation by *Saccharomyces cerevisiae*. *Green Chem.* **18**, 4657–4666 (2016).
4. Galinato, M. G. I., Niedzwiedzki, D., Deal, C., Birge, R. R. & Frank, H. A. Cation radicals of xanthophylls. *Photosynth. Res.* **94**, 67–78 (2007).
5. Ceroni, F. et al. Burden-driven feedback control of gene expression. *Nat. Methods.* **15**, 387–393 (2018).

Reviewers' Comments:

Reviewer #1:

Remarks to the Author:

I thank the authors for considering my points, performing additional experiments, and improving their manuscript. In particular, I think the results are more interesting now that the feedback promoter strain performs far better than the cra knockout. All of my points were fully addressed and I am happy for the manuscript to be published.

Reviewer #2:

Remarks to the Author:

Feedback-regulated promoters enable robust control of synthetic metabolic pathways by stabilizing metabolite levels

Chun-Ying Wang, Martin Lempp, Niklas Farke, Stefano Donati, Timo Glatter, Hannes Link

Summary

The additional experiments and revisions to the manuscript have improved the clarity and interpretation of the data and findings in some respects, but have also raised new concerns. It has become clearer that the study does not demonstrate, and would not be able to demonstrate without a completely new series of experiments, that negative feedback by Cra on the regulation of the pBAD promoter has an effect that is distinguishable from a weaker pBAD promoter (point 8 below). In addition, the claims of robustness and stability are unclear because they are not well defined, and they do not appear to be supported by the data (points 1 and 2 below). There are three main findings of the manuscript (a) overproduction of glycerol production decreases growth in *E. coli* (Fig. 1), (b) incorporating a binding site for the Cra transcription factor into the pBAD promoter can overcome the growth effect (Fig. 3 and 4) and (c) the above could potentially enable bioengineers to regulate the expression of a wide range of synthetic pathways that use glycolytic metabolites (Fig. 6). It is not clear that point (b) specifically requires feedback regulation, and that the same effect couldn't be achieved with the common practice of using a weaker promoter instead of pBAD, which reduces the impact of point (c) and the novelty of this manuscript.

1. Removing the claim of ultrasensitivity has improved the manuscript. However, the authors have brought to the forefront the issue of what they mean by robustness with the claim of stabilization in the title and the new analysis of robustness (Fig. 3b). Robustness was not defined in the original manuscript but I interpreted it as meaning less decrease in glycerol production with the induction of pBAD by arabinose (based on the abstract and Fig 6 caption). I now realize the original manuscript had a second meaning, which is the growth rate is reduced less by the induction of pBAD (Fig. 4 in original and revised manuscripts). In the revised manuscript, there is a third meaning for robustness. In Fig. 3b and the associated analysis, robustness is the capacity of the system to be stable (steady state or stable oscillation) at different parameter values. I have technical questions about the analysis below. There is a fourth meaning to robustness, which is mentioned in the last sentence of the Discussion, and that is it can refer to genetic noise or the amount of variance in protein production. Robustness must be precisely defined and used consistently. The above meanings are very different and need to be measured and analyzed differently. Please also see point 8.

2. Similar to the concerns about the use of the term robustness, the authors refer to "stabilizing metabolite levels" in the title. What do the authors mean by stabilizing? The terms stable, stabilizing and other variants are used in imprecise ways throughout the manuscript. For the reasons explained in this section and the one above, it is not clear what is meant in the title by "robust control of synthetic metabolic pathways by stabilizing metabolite levels."

3. The new robustness analysis (shown in Fig. 3b) is not supported by any experiments and its connection to the rest of the study is not obvious. As stated in the citation, EMRA is designed to determine "the effects of parameter drifting until a bifurcation point, beyond which a stable steady state disappears and system failure occurs." The use of this analysis in this study is not appropriate. There was no evidence presented of a bifurcation point, EMRA analysis is not directly

related to assessment of the sensitivity of production levels, growth rate or noise in pBAD induction, and there is no experimental data to indicate whether the analysis is accurate or meaningful for this study. There are questions about the way the EMRA was conducted, including how and why the authors solved for the steady state (instead of equilibrium), whether the random sampling took into account that α_1 and α_2 are in the exponent, and the equations that were analyzed (see next point). The theoretical analyses need to be connected to the experiments to be valuable. What does this analysis tell us about protein production levels or growth rate that can be confirmed?

4. The authors have not addressed the original concerns about the ODE model. In equation 1, $d\text{FBP}/dt$ has two terms, $r_{\text{lower_glycolysis}}$ and r_{glycerol} , which are described in Eq. 3 and 4. Both of these equations use the Michaelis-Menten equation with FBP in both the numerator and denominator. FBP does not occur elsewhere in Eq. 3 and 4, therefore the solution to eq. 3 and 4 does not have units of nM of FBP, and this is a problem because they are independent of the FBP concentration. Consider the scenario where $\text{FBP} = 0$ nM. In this scenario, according to Eq. 1 there will be degradation of FBP molecules (at a rate of $k_{\text{cat}1} \cdot e_1 \cdot 1/k_{\text{m}1} + k_{\text{cat}2} \cdot e_2 \cdot 1/k_{\text{m}2}$) despite there being no FBP molecules in the cell. Clearly this does not make sense, and it can also generate negative concentrations of FBP. Negative concentrations may not have been obvious in the simulations because the parameters were constrained so that the sum of Eq. 3 and 4 < 66.67 mM min^{-1} at steady state. Nonetheless it is a poorly constructed model and should be changed. The default expectation should be that the clearance of the FBP will be proportional to its concentration unless there is evidence to the contrary (such as experimental evidence of zero order kinetics). It is not appropriate to use the standard Michaelis-Menten equation without modification because the Michaelis-Menten equation assumes concentrations of substrate (FBP) that are far in excess of the enzyme concentration, which is not the case in this system.

5. The authors have partially addressed the concern about the accuracy of counting cell number by OD measurements alone, by measuring the side scatter and forward scatter in flow cytometry to estimate cell size. However, further analysis is required. It is not possible to determine whether there is a difference in cell size by only inspecting the scatterplots because it is difficult to see by eye small but meaningful differences in the distribution on the large range of the log scale. There are papers describing software that can help with this analysis. At a minimum the authors should show the histogram of the forward and side scatters and indicate the mean and median of the distributions. Alternatively, the authors can count the colonies to convert OD to CFUs.

6. There is a section titled: "A metabolic model predicts optimal control of the glycerol pathway". There is no explanation of what optimal control means or what is specifically being "optimized". This requires an explanation, or simpler language to replace optimal control that better describes what was done and found.

7. The y-axis label in Fig 1b of the original manuscript was the protein amount (GFP/OD), but in the revised manuscript it is GFP expression ($d\text{GFP}/dt/\text{OD}$). Why was that change made? It is also unclear what the new measurement is. Based on the units and description in the text it is measuring the derivative determined by measurements of GFP expression at time points between 7 and 9 hours. But it is called promoter expression, which is not the same as $d\text{GFP}/dt$. $d\text{GFP}/dt$ depends on protein clearance as well as production, and protein clearance depends on cell growth. Cell growth is not constant especially at these time points after mid-exponential phase (Fig S2).

8. The manuscript states that the weaker promoter is less robust (this meaning is unclear as mentioned above) than the pBAD-Cra strain and that this indicates that feedback regulation by Cra increased robustness of the pBAD-Cra strain rather than lower promoter activity per se. I did not understand how that the data supports that perspective. In fact, it can be interpreted to show the opposite. The pBAD promoter has a normalized activity level of 1 and the pBAD-Cra promoter has an activity level of 0.26 (3.8 fold lower expression). The weaker promoter has an intermediate level of activity of 0.5 (2 fold lower) and therefore would be expected to display a growth rate and glycerol production that is between the pBAD-Cra and pBAD promoters, and it does (compare Fig. 4a and lower row of Fig. 4b). Interestingly the delta Cra strain has very similar growth rates and glycerol production as the weaker promoter (Fig. 3). It was unclear whether the delta Cra strain had the pBAD or the pBAD-Cra promoter. In the former case, the insertion of the cra sequence in

pBAD is primarily having an effect on the transcription of the promoter. In the latter case, the insertion of the *cra* sequence in pBAD is having a direct effect on the transcription of the promoter as well as enabling repression by Cra. In either case, the *cra* sequence in pBAD is directly modifying transcription levels. On a related note, the absence of any effect of the insertion of binding sequences for other transcription factors (Fig. 4c) is difficult to interpret without seeing their effect on pBAD transcription, which could be evaluated with GFP, and demonstrating that these sites are functional with a positive control.

The claim that only feedback by Cra generates improves robustness and not lower promoter activity per se, raised the question of what properties we would expect to see due to feedback that are not explained by lower promoter activity.

Negative feedback has several well-established effects in biological systems. It aids homeostasis by decreasing variation around a set point and can help systems achieve the set point faster, it can act as a limiter to prevent levels of a factor exceeding a maximum level, and it can help create oscillations (<https://pubmed.ncbi.nlm.nih.gov/18927383/>). Negative feedback can also be a component in regulatory systems to generate other complex behaviors. In this study, the only property of negative feedback that was examined is the role in limiting transcription from the pBAD promoter and this can be achieved by other means such as weakening the pBAD promoter. In other words, the experiments have not looked at any behaviors or properties that are specific to negative feedback and therefore why Cra feedback regulation is required to improve the output of this system. To demonstrate that negative feedback is the preferred solution over a weaker promoter requires new dynamics measurements or measurements of genetic noise.

Response to Reviewer #2

Summary

The additional experiments and revisions to the manuscript have improved the clarity and interpretation of the data and findings in some respects, but have also raised new concerns. It has become clearer that the study does not demonstrate, and would not be able to demonstrate without a completely new series of experiments, that negative feedback by Cra on the regulation of the pBAD promoter has an effect that is distinguishable from a weaker pBAD promoter (point 8 below). In addition, the claims of robustness and stability are unclear because they are not well defined, and they do not appear to be supported by the data (points 1 and 2 below). There are three main findings of the manuscript (a) overproduction of glycerol production decreases growth in *E. coli* (Fig. 1), (b) incorporating a binding site for the Cra transcription factor into the pBAD promoter can overcome the growth effect (Fig. 3 and 4) and (c) the above could potentially enable bioengineers to regulate the expression of a wide range of synthetic pathways that use glycolytic metabolites (Fig. 6). It is not clear that point (b) specifically requires feedback regulation, and that the same effect couldn't be achieved with the common practice of using a weaker promoter instead of pBAD, which reduces the impact of point (c) and the novelty of this manuscript.

We agree with most points of this reviewer and appreciate the time and thoughts he has given to our work. We addressed all points in this revision and believe they improved the manuscript. An important revision is that we removed all claims about robustness, stability and dynamic feedback. Instead we focus on our observations, as described in points a), b) and c) below. However we respectfully disagree with some concerns about the model and the way we conducted the model analysis (a detailed response is below).

Overall, we agree with the reviewer's interpretations of our main findings:

a) **"Overproduction of glycerol decreases growth."** The important point here is that growth decreases due to a regulatory problem, which is caused by Cra. We provide 3 separate lines of evidence for this effect: i) growth/metabolome/proteome data in Figure 1 and 2, ii) growth of the Δ cra strain in Figure 2, and iii) the model analysis in Figure 3. Thus a large part of the manuscript describes this finding, and we decided to put a stronger focus on it. For example by changing the title:

"Metabolome and proteome analyses of glycerol overproducing *E. coli* unravel transcriptional misregulation of glycolysis and guide optimization strategies"

b) **"Incorporating a binding site for the Cra transcription factor into the pBAD promoter overcomes the growth effect"**. We agree with the reviewers concern that it's difficult to distinguish dynamic regulation by Cra from a static effect. In other words, it is possible that Cra simply inhibits the promoter, which is similar to constructing a weaker promoter. But we don't understand why this would change the impact of the finding, because inhibition by Cra has several advantages over constructing a weaker promoter (e.g. the "right" strength is difficult to find). We described this in detail in our response to point 8.

c) **“The approach might enable bioengineers to regulate the expression of a wide range of synthetic pathways that use glycolytic metabolites (Fig. 6).”** We agree and want to emphasize that we also show that the approach applies to a **wide range of promoters**. In all cases the Cra regulated promoter performs best, which already indicates that Cra regulation automatically reduces promoter activity by the “right” level (otherwise one would have to screen many variants of weaker promoters).

1. Removing the claim of ultrasensitivity has improved the manuscript. However, the authors have brought to the forefront the issue of what they mean by robustness with the claim of stabilization in the title and the new analysis of robustness (Fig. 3b). Robustness was not defined in the original manuscript but I interpreted it as meaning less decrease in glycerol production with the induction of pBAD by arabinose (based on the abstract and Fig 6 caption). I now realize the original manuscript had a second meaning, which is the growth rate is reduced less by the induction of pBAD (Fig. 4 in original and revised manuscripts). In the revised manuscript, there is a third meaning for robustness. In Fig. 3b and the associated analysis, robustness is the capacity of the system to be stable (steady state or stable oscillation) at different parameter values. I have technical questions about the analysis below. There is a fourth meaning to robustness, which is mentioned in the last sentence of the Discussion, and that is it can refer to genetic noise or the amount of variance in protein production. Robustness must be precisely defined and used consistently. The above meanings are very different and need to be measured and analyzed differently. Please also see point 8.

We agree with the concern that robustness and stabilization were not unequivocally defined. Therefore we removed all statements about robustness and stabilization (except the model analysis). Instead we simply state what we observe:

Higher growth rates at higher glycerol production rates.

To emphasize this we performed additional experiments in which we measured glycerol production rates in the base strain and the pBAD-Cra strain. The results are shown in a new Figure 1e (for the base strain), and in a new Figure 3f (for the pBAD-Cra strain). The results quantitatively confirm that the pBAD-Cra strain achieves higher growth rates at higher glycerol production rates. We changed this throughout the manuscript and put a focus on the relationship between glycerol production rates and growth rates.

2. Similar to the concerns about the use of the term robustness, the authors refer to “stabilizing metabolite levels” in the title. What do the authors mean by stabilizing? The terms stable, stabilizing and other variants are used in imprecise ways throughout the manuscript. For the reasons explained in this section and the one above, it is not clear what is meant in the title by “robust control of synthetic metabolic pathways by stabilizing metabolite levels.”

We agree with this point and removed the term stabilizing metabolite levels throughout the revised manuscript. We also changed the title. Instead we describe the observation that the pBAD-Cra strain achieves higher FBP levels at higher glycerol fluxes, than the base strain. We also believe that this indicates that the FBP-Cra interaction improves the pBAD-Cra strain. We describe this in the revised text:

Results:

“Thus, the pBAD-Cra strain can maintain higher FBP levels at higher glycerol production rates than the base strain. This suggested that the interaction between FBP and Cra, in combination with the pBAD promoter, counteracts decreases of FBP: i) if FBP falls below a critical value, Cra activity increases, ii) higher Cra activity represses the pBAD-Cra promoter and decreases GPD1 expression, and iii) lower expression of GPD1 will restore the concentration of FBP.”

Discussion:

“Here we show that engineering transcriptional regulation (Cra-regulation) in a synthetic glycerol pathway can help to maintain regulatory metabolites above a critical threshold: the pBAD-Cra strain has higher FBP levels and at the same time a higher glycerol flux than the base strain. This supports the hypothesis that Cra-dependent regulation counteracts a dramatic decline in the concentration of FBP by downregulating the expression of the glycerol pathway in response to decreasing FBP levels.”

3. The new robustness analysis (shown in Fig. 3b) is not supported by any experiments and its connection to the rest of the study is not obvious. As stated in the citation, EMRA is designed to determine “the effects of parameter drifting until a bifurcation point, beyond which a stable steady state disappears and system failure occurs.” The use of this analysis in this study is not appropriate. There was no evidence presented of a bifurcation point, EMRA analysis is not directly related to assessment of the sensitivity of production levels, growth rate or noise in pBAD induction, and there is no experimental data to indicate whether the analysis is accurate or meaningful for this study. There are questions about the way the EMRA was conducted, including how and why the authors solved for the steady state (instead of equilibrium), whether the random sampling took into account that α_1 and α_2 are in the exponent, and the equations that were analyzed (see next point).

The theoretical analyses need to be connected to the experiments to be valuable. What does this analysis tell us about protein production levels or growth rate that can be confirmed?

We agree that robustness was difficult to relate to the experiments and we revised the model analysis. We kept the same EMRA analysis, but additionally show the **maximal glycerol production rate** that can be achieved with each model. Therefore we evaluated for each parameter set and each model the maximal glycerol flux. The results are shown in a new Figure 3.

These model results match our experimental observation: the 2xcra model has the highest glycerol production rates, then the Δcra model, and the base strain model is the worst. We base this conclusion on the fraction of parameters that achieve high glycerol production rates. The logic is that the more parameters achieve a high flux; the more likely it is that the real system achieves high fluxes. This is a standard assumption in all ensemble model approaches (also for models used in the weather forecast).

Thus we believe the EMRA analysis is an important result in our work, because it allows us to gradually increase induction of enzyme 2 and test the resulting consequences. We performed this analysis in the same way as it was described in the original work of Lee et al. [Ref 23]. Lee et al. varied maximal rates of enzymes (V_{max}) with the continuation method and then tested for bifurcation points. We increased the

maximal rate of e2 expression (β_2) and then tested for bifurcation points. A bifurcation point occurs when the Jacobian matrix becomes singular (an eigenvalue approaches or passes 0). At this point, the system becomes unstable which can be because of metabolite accumulation/depletion or oscillations (but not due to negative concentrations, as the reviewer suspects).

We don't understand what the reviewer refers to with "equilibrium" and how we could solve the ODE model for equilibrium. We always solve for steady states, which means reaction rates fulfill mass balances: influx of glucose equals glycerol flux plus flux in lower glycolysis. We also want to emphasize that concentrations are never negative in our model (see next point).

4. The authors have not addressed the original concerns about the ODE model. In equation 1, $d\text{FBP}/dt$ has two terms, $r_{\text{lower_glycolysis}}$ and r_{glycerol} , which are described in Eq. 3 and 4. Both of these equations use the Michaelis-Menten equation with FBP in both the numerator and denominator. FBP does not occur elsewhere in Eq. 3 and 4, therefore the solution to eq. 3 and 4 does not have units of nM of FBP, and this is a problem because they are independent of the FBP concentration. Consider the scenario where $\text{FBP} = 0$ nM. In this scenario, according to Eq. 1 there will be degradation of FBP molecules (at a rate of $k_{\text{cat}1} \cdot e_1 / (k_{\text{m}1} + k_{\text{cat}2} \cdot e_2)$) despite there being no FBP molecules in the cell. Clearly this does not make sense, and it can also generate negative concentrations of FBP. Negative concentrations may not have been obvious in the simulations because the parameters were constrained so that the sum of Eq. 3 and 4 < 66.67 mM min^{-1} at steady state. Nonetheless it is a poorly constructed model and should be changed. The default expectation should be that the clearance of the FBP will be proportional to its concentration unless there is evidence to the contrary (such as experimental evidence of zero order kinetics). It is not appropriate to use the standard Michaelis-Menten equation without modification because the Michaelis-Menten equation assumes concentrations of substrate (FBP) that are far in excess of the enzyme concentration, which is not the case in this system.

We respectfully disagree with this point, because our Equations 3 and 4 are correct and follow the Michaelis-Menten Equation as it is described in biochemistry textbooks.

The statements above are wrong, because when FBP is zero, the rates are also zero:

Correct is: $d\text{FBP}/dt = 0 = k_{\text{cat}1} \cdot e_1 / (0 + k_{\text{m}1}) + k_{\text{cat}2} \cdot e_2 / (0 + k_{\text{m}2})$

The reviewer's rearrangement of equation 1 is wrong: **$k_{\text{cat}1} \cdot e_1 / k_{\text{m}1} + k_{\text{cat}2} \cdot e_2 / k_{\text{m}2}$**

Negative concentrations cannot occur in our model, because all rates are zero at zero concentrations of FBP, or of the enzyme. Also the unit is correct; reaction rates are **mmol/s/L**. Because:

- k_{cat} has (1/s),
- e has (mmol/L)
- the term $\text{FBP}/(\text{FBP} + K_{\text{m}})$ has no unit; (mmol/L) / (mmol/L)

We also disagree with the following statement of the reviewer:

“It is not appropriate to use the standard Michaelis-Menten equation without modification because the Michaelis-Menten equation assumes concentrations of substrate (FBP) that are far in excess of the enzyme concentration, which is not the case in this system.”

The Michaelis-Menten Equation is valid also at low metabolite concentrations. The Michaelis-Menten Equation is the gold standard for ODE models of metabolic pathways and it is widely used in the modelling community (described for example here, doi: 10.1186/1742-4682-3-41).

5. The authors have partially addressed the concern about the accuracy of counting cell number by OD measurements alone, by measuring the side scatter and forward scatter in flow cytometry to estimate cell size. However, further analysis is required. It is not possible to determine whether there is a difference in cell size by only inspecting the scatterplots because it is difficult to see by eye small but meaningful differences in the distribution on the large range of the log scale. There are papers describing software that can help with this analysis. At a minimum the authors should show the histogram of the forward and side scatters and indicate the mean and median of the distributions. Alternatively, the authors can count the colonies to convert OD to CFUs.

We replaced the scatterplots with histograms of the forward and side scatters and marked the mean of the distributions (in Supplementary Figure. 7). The result showed that the cell size reduces only in the base strain with 0.5% ara, confirming that this strain does not perform well at high ara induction also with respect to this parameter.

New Supplementary Figure. 7

6. There is a section titled: “A metabolic model predicts optimal control of the glycerol pathway”. There is no explanation of what optimal control means or what is specifically being “optimized”. This requires an explanation, or simpler language to replace optimal control that better describes what was done and found.

We agree and used a section title that better describes the purpose of the model:

“A metabolic model predicts optimization strategies for glycerol production”

7. The y-axis label in Fig 1b of the original manuscript was the protein amount (GFP/OD), but in the revised manuscript it is GFP expression (dGFP/dt/OD). Why was that change made? It is also unclear what the new measurement is. Based on the units and description in the text it is measuring the derivative determined by measurements of GFP expression at time points between 7 and 9 hours. But it is called promoter expression, which is not the same as dGFP/dt. dGFP/dt depends on protein clearance as well as production, and protein clearance depends on cell growth. Cell growth is not constant especially at these time points after mid-exponential phase (Fig S2).

We decided to show dGFP/dt/OD because it is a better description of the promoter activity than protein amount (GFP/OD). This method was introduced by the Uri Alon Lab and we performed all calculations as described in Zaslaver et al. (2008) *Nature Methods* (doi: 10.1038/nmeth895.)

8. The manuscript states that the weaker promoter is less robust (this meaning is unclear as mentioned above) than the pBAD-Cra strain and that this indicates that feedback regulation by Cra increased robustness of the pBAD-Cra strain rather than lower promoter activity per se. I did not understand how that the data supports that perspective. In fact, it can be interpreted to show the opposite. The pBAD promoter has a normalized activity level of 1 and the pBAD-Cra promoter has an activity level of 0.26 (3.8 fold lower expression). The weaker promoter has an intermediate level of activity of 0.5 (2 fold lower) and therefore would be expected to display a growth rate and glycerol production that is between the pBAD-Cra and pBAD promoters, and it does (compare Fig. 4a and lower row of Fig. 4b).

Overall, we agree with the reviewer’s argument that it is difficult to distinguish negative feedback by Cra from the effects of a weaker pBAD promoter. It is indeed likely that a weaker promoter with the right strength performs equally well as the pBAD-Cra promoter, but finding that promoter would be much more difficult and is probably different for different conditions. We describe this in the revised text:

“Thus, the pBAD-Cra promoter is a 3.7-fold weaker version of the pBAD promoter, probably because Cra represses the promoter. In principle the same effect could be achieved by other mutations that decrease activity of the pBAD promoter, for example mutations between the -10 and -35 boxes²⁴. Therefore, we constructed a pBAD promoter with mutations between the -10 and -35 boxes that decrease activity by a factor of two²⁴, and analyzed the resulting pBAD-weak strain. The pBAD-weak strain grew indeed better than the base strain and achieved higher glycerol titers (Fig. 4a). However, the pBAD-weak strain performed worse than the pBAD-Cra strain, which might be due to the different activities of the pBAD-Cra and the pBAD-weak promoter (3.7-fold and 2-fold lower activity than the original pBAD promoter, respectively). Another explanation for the better growth of the pBAD-Cra strain is that the pBAD-Cra promoter is dynamic and the pBAD-weak promoter is static.”

Interestingly the delta Cra strain has very similar growth rates and glycerol production as the weaker promoter (Fig. 3). It was unclear whether the delta Cra strain had the pBAD or the pBAD-Cra promoter. In the former case, the insertion of the cra sequence in pBAD is primarily having an effect on the transcription of the promoter. In the latter case, the insertion of the cra sequence in pBAD is having a direct effect on the transcription of the promoter as well as enabling repression by Cra. In either case, the cra sequence in pBAD is directly modifying transcription levels.

The strain in Fig. 3c in the manuscript was the delta Cra strain with the pBAD promoter. It does not allow conclusion about the consequences of inserting the Cra sequence. Instead it shows that the deleting Cra improves growth and glycerol production. This in turn confirms our hypothesis about misregulation by Cra.

On a related note, the absence of any effect of the insertion of binding sequences for other transcription factors (Fig. 4c) is difficult to interpret without seeing their effect on pBAD transcription, which could be evaluated with GFP, and demonstrating that these sites are functional with a positive control.

We performed the suggested experiments and the results are shown as Figure R1. Although we inserted binding sites of repressors, the engineered promoters showed higher GFP expression relative the original pBAD promoter. Thus, regulation of these promoters is not functional. This in turn means that the results in the previous Figure 4c are not conclusive and we decided to remove them. We thank the reviewer for suggesting this important control.

Figure R1. Binding sites of different transcription factors were inserted into the pBAD promoter and GFP was expressed with these promoters. The promoter activity is shown for different arabinose concentrations. Only insertion of Cra reduced promoter activity compared to the pBAD Cra promoter, suggesting that only the pBAD-Cra promoter is functional.

The only promoter that was functional was the pBAD-Cra promoter, because it showed lower GFP expression than the pBAD promoter. To obtain additional evidence that the pBAD-Cra promoter is truly regulated by Cra we expressed GFP with the pBAD and the pBAD-Cra promoter in the Cra deletion strain. In the Δ cra strain GFP was even slightly higher with the pBAD-Cra promoter than with the pBAD promoter, confirming that Cra inhibits the promoter and that the pBAD-Cra promoter is functional. We added this as a new Figure 4c and in the text:

“To obtain further evidence that the pBAD-Cra promoter is functional, we measured activity of the pBAD promoter and the pBAD-Cra promoter with GFP, both in the wild-type and the Δ cra strain. In the wild-

type, the pBAD-Cra promoter had a lower activity than the pBAD promoter (Fig. 4c), thus indicating that Cra inhibits the promoter. In the Δ cra strain, however, the pBAD-Cra promoter had a slightly higher activity than the pBAD promoter (Fig. 4c). These results show that the pBAD-Cra promoter is functional: Cra inhibits the pBAD-Cra promoter and this regulation is missing in the absence of Cra (in the Δ cra strain). Thus the lower activity of the pBAD-Cra is not merely due sequence changes but due to active inhibition by Cra.”

The claim that only feedback by Cra generates improves robustness and not lower promoter activity per se, raised the question of what properties we would expect to see due to feedback that are not explained by lower promoter activity. Negative feedback has several well-established effects in biological systems. It aids homeostasis by decreasing variation around a set point and can help systems achieve the set point faster, it can act as a limiter to prevent levels of a factor exceeding a maximum level, and it can help create oscillations (<https://pubmed.ncbi.nlm.nih.gov/18927383/>). Negative feedback can also be a component in regulatory systems to generate other complex behaviors. In this study, the only property of negative feedback that was examined is the role in limiting transcription from the pBAD promoter and this can be achieved by other means such as weakening the pBAD promoter. In other words, the experiments have not looked at any behaviors or properties that are specific to negative feedback and therefore why Cra feedback regulation is required to improve the output of this system. To demonstrate that negative feedback is the preferred solution over a weaker promoter requires new dynamics measurements or measurements of genetic noise.

We want to emphasize that previous work on engineered feedback in synthetic metabolic merely showed that it improves productivity (see ref 11-15). We believe that our work goes one step further by measuring the effects at the molecular level, especially FBP levels. But we also agree that future work must consider effects like set-point variation, response times, etc. We emphasize this in our discussion and added the reference suggested by the reviewer (ref 25).

“Thus our data indicate that this feedback regulation is functional, because it enabled high FBP levels and at the same time high glycerol production rates, which presumably prevented that E. coli switches from glycolysis to gluconeogenesis. However, further experiments are required to experimentally investigate how this feedback loop shapes metabolism in space and time²⁵.”

We also agree that a weaker promoter (with the right strength) would perform similar than the Cra regulated variant. But that does not mean the pBA-Cra promoter is not dynamic. It is likely that in different environments the promoter would automatically adapt to the right strength, while the weaker promoter must be constructed from scratch. We mention this in the revised discussion:

“However, as it remains an open question whether this regulation is truly dynamic, we cannot rule out the possibility that, due to the constant inhibitory activity of Cra, the pBAD-Cra promoter simply functions as a weaker pBAD promoter. Future studies should clarify whether the pBAD-Cra promoter automatically adapts to new conditions, e.g. by shifting the glycerol producers between different environments.”

Reviewers' Comments:

Reviewer #2:

Remarks to the Author:

Metabolome and proteome analyses of glycerol overproducing *E. coli* unravel transcriptional misregulation of glycolysis and guide optimization strategies

Chun-Ying Wang, Martin Lempp, Niklas Farke, Stefano Donati, Timo Glatter, Hannes Link

Summary

The authors have not addressed some of the key issues that were raised in previous reviews. One issue is that the ODE model and a second issue is that it is difficult from the limited information provided for the EMRA section to determine whether it was done correctly and what the results mean. I think these points will be apparent to subject matter experts but not for most readers. The model and EMRA are not required for the conclusions, and therefore the manuscript could be published without them.

Major points

1. As the authors noted, the equation I wrote in the previous review to try and explain the issue with the Michaelis-Menten Equation in Equation 1 was incorrect. But the underlying issue with Equation 1 remains.

a. $d\text{FBP}/dt$ has three terms. The first term, $r_{\text{upper_glycolysis}}$ has an "influx rate", which is 66.67 mM/min which "corresponds" to 8 mmol/g/h. It was not explained how this rate for glucose was converted to a production rate of FBP. The first term should have units of mM min^{-1} of FBP. The second term, $r_{\text{lower_glycolysis}}$, is the product of $k_{\text{cat},1}$, which has units of min^{-1} , e_1 , which has units of mM of enzyme 1, and $\text{FBP}/(\text{FBP}+K_{\text{m}2})$, which is unitless. Therefore, units for the second term are mM min^{-1} of e_1 . The third term, r_{glycerol} , is the product of $k_{\text{cat},2}$, which has units of min^{-1} , e_2 , which has units of mM of enzyme 2, and $\text{FBP}/(\text{FBP}+K_{\text{m}2})$, which is unitless. Therefore, units for the third term are mM min^{-1} of e_2 . Therefore Equation 1, which is rate equation for FBP, appears to be subtracting rates for different chemical species – nM min^{-1} of FBP, nM min^{-1} of e_1 and nM min^{-1} of e_2 . While FBP (or glucose?), enzyme 1 and enzyme 2 all have units of mM min^{-1} , they can't be added or subtracted because they are different chemical species, and this equation is modeling the rate of FBP. This is the equivalent of subtracting 4 iron atoms per liter from 10 carbon atoms per liter and saying there are 6 atoms of carbon per liter. Each term in the equation should only have units of nM min^{-1} of FBP.

b. The authors state in their rebuttal "We respectfully disagree with this point, because our Equations 3 and 4 are correct and follow the Michaelis-Menten Equation as it is described in biochemistry textbooks " and "The Michaelis-Menten Equation is valid also at low metabolite concentrations. The Michaelis-Menten Equation is the gold standard for ODE models of metabolic pathways and it is widely used in the modelling community (described for example here, doi: 10.1186/1742-4682-3-41)."

i. The point is not whether the Michaelis-Menten Equation (MME) is the "gold standard" (and it isn't in this case), but whether it is being used appropriately in the model. For the reasons listed below, I don't think it is appropriate.

ii. MME is not typically used to model degradation (MME in the production term is fine) because the removal of most chemical species is usually considered a first order reaction. If it is treated as zero order (which should be explained) it should still have units of nM min^{-1} of FBP. If the authors wish to use MME in Eq. 1 they need to explain why. One interpretation of Eq. 1 is that the authors are stating that the clearance of FBP by the enzymes is dependent on the function $\text{FBP}/(\text{FBP}+K)$. What does this mean? Because this is dimensionless, does this portion of the equation describe the probability of clearance for each molecule of FBP at a given concentration of FBP for each enzyme molecule. A single molecule of FBP will have a lower probability of clearance by an enzyme when the FBP concentration is low and a high probability of clearance by an enzyme when the FBP concentration is high. This is the opposite of what is expected because at high concentrations you would expect competition between FBP molecules for the enzyme binding site. I recommend the reference in section 1d in this review for more information on the MME and Chapter 2 in An Introduction to Systems Biology (Design Principles of Biological Circuits) by U. Alon for modeling molecule clearance.

iii. MME is usually used to simplify the modeling of chemical kinetics by making certain assumptions such as the total enzyme concentration is constant, and the substrate concentrations

>> enzyme concentration. However, the model is using MME without respecting these assumptions and that is potentially problematic. The authors declared that MME is valid at low metabolite concentration without providing a source. Here are two textbooks and even a Wikipedia entry with clear statements contradicting that declaration.

1. "That the substrate concentration is much larger than the enzyme concentration." Page 43 in

<https://link.springer.com/content/pdf/10.1007%2F978-1-349-01959-5.pdf>

2. "An essential feature of Michaelis–Menten kinetics is that the catalyst becomes saturated at high substrate concentrations."

<https://www.sciencedirect.com/topics/engineering/michaelis-menten-equation>

3. "Under certain assumptions – such as the enzyme concentration being much less than the substrate concentration – the rate of product formation is given by

https://en.wikipedia.org/wiki/Michaelis%E2%80%93Menten_kinetics

c. It is not clear why simple mass action kinetics was not used instead of MME because the predictions are simulated (as opposed to an analytical solution), and it would be very feasible to add a few more equations to the model. This approach is more standard because it explicitly models each reaction step and each species. If the model must be included in the model, why not explicitly model the enzyme-substrate complex and the degradation of the substrate? Then the model would be applicable for all the conditions that are being modeled, including dynamically varying enzyme concentration, and there would be no concern about any of the assumptions of the MME, including a constant total enzyme conc, product formation being slower than association and dissociation of the enzyme-substrate complex, and the enzyme conc being less than the substrate conc.

d. I refer the authors to Chapter 6 (Modeling Molecular Interaction Networks with Nonlinear Ordinary Differential Equations by Emery D. Conrad and John J. Tyson) in System Modeling in Cellular Biology: From Concepts to Nuts and Bolts (Eds: Zoltan Szallasi, Jorg Stelling, and Vipul Periwa) on the application of MME and mass action kinetic models.

2. EMRA.

a. It is unclear that the experiments match the predictions for the 2xcra model. The axes on the plots on Fig. 3c (model) and 3f are different so it is not possible for the reader to directly compare them, and therefore to draw any conclusion as to whether the experiments do or do not agree with the predictions.

b. The comment in the previous review on equilibrium vs steady state, was one of several general questions posed to prompt the authors to provide more details about the analysis. For example, have the authors considered cases where the eigenvalue had no zero real parts but a complex part, such as stable oscillations. This is an important consideration in a system with feedback. Following from the above, the manuscript states for the analysis "1st criterion: all real parts of the eigenvalues of the system's Jacobian need to be negative." "Once an eigenvalue reaches zero, the Jacobian becomes singular and matrix inversion is no longer possible. This bifurcation point defines the boundary between the stable and unstable parameter space. In other words: after this point is passed, the system can't return to a stable steady state." Should the cut-off be eigenvalues >0 to permit stable oscillations?

c. Because the EMRA is based on the ODE model, and it has the issues described above in comment 1, I don't have confidence in the predictions of the EMRA model.

Third revision and appeal by the Authors

Our response is **red**

References are **blue**

Summary

The authors have not addressed some of the key issues that were raised in previous reviews. One issue is that the ODE model and a second issue is that it is difficult from the limited information provided for the EMRA section to determine whether it was done correctly and what the results mean. I think these points will be apparent to subject matter experts but not for most readers. The model and EMRA are not required for the conclusions, and therefore the manuscript could be published without them.

Major points

1. As the authors noted, the equation I wrote in the previous review to try and explain the issue with the Michaelis-Menten Equation in Equation 1 was incorrect. But the underlying issue with Equation 1 remains.

a. $d\text{FBP}/dt$ has three terms. The first term, $r_{\text{upper_glycolysis}}$ has an “influx rate”, which is 66.67 mM/min which “corresponds” to 8 mmol/g/h. It was not explained how this rate for glucose was converted to a production rate of FBP. The first term should have units of mM min^{-1} of FBP. The second term, $r_{\text{lower_glycolysis}}$, is the product of $k_{\text{cat},1}$, which has units of min^{-1} , e_1 , which has units of mM of enzyme 1, and $\text{FBP}/(\text{FBP}+K_{m2})$, which is unitless. Therefore, units for the second term are mM min^{-1} of e_1 . The third term, r_{glycerol} , is the product of $k_{\text{cat},2}$, which has units of min^{-1} , e_2 , which has units of mM of enzyme 2, and $\text{FBP}/(\text{FBP}+K_{m2})$, which is unitless. Therefore, units for the third term are mM min^{-1} of e_2 . Therefore Equation 1, which is rate equation for FBP, appears to be subtracting rates for different chemical species – nM min^{-1} of FBP, nM min^{-1} of e_1 and nM min^{-1} of e_2 . While FBP (or glucose?), enzyme 1 and enzyme 2 all have units of mM min^{-1} , they can't be added or subtracted because they are different chemical species, and this equation is modeling the rate of FBP. This is the equivalent of subtracting 4 iron atoms per liter from 10 carbon atoms per liter and saying there are 6 atoms of carbon per liter. Each term in the equation should only have units of nM min^{-1} of FBP.

Point 1a is not correct: we do subtract the same chemical species. The turnover number k_{cat} of an enzyme is defined as the number of substrate molecules transformed per time by a single enzyme molecule (Ref 1 and 2).

In more detail, k_{cat} is defined as:

a) $k_{\text{cat}} = V_{\text{max}} / E$ which has the formal unit $[\text{mM}_{\text{met}} * \text{min}^{-1} * \text{mM}_E^{-1}]$

Michaelis Menten kinetics give the rate of an enzymatic reaction as:

$$b) \quad v = V_{\max} * [S]/([S]+K_m)$$

Correspondingly, we can substitute V_{\max} with $E * k_{\text{cat}}$.

$$c) \quad v = E * k_{\text{cat}} * [S]/([S]+K_m)$$

Which is precisely Eq 3) and Eq 4) in our original manuscript, and therefore they have the units $\text{mM}(\text{metabolite}) * \text{min}^{-1}$ and not $\text{mM}(\text{enzyme}) * \text{min}^{-1}$ as suspected by the reviewer.

Reference 1: "... the turnover number of an enzyme, which is the number of **substrate molecules converted into product by an enzyme molecule in a unit time** when the enzyme is fully saturated with substrate. It is equal to the kinetic constant k_2 , which is also called k_{cat} "

Berg JM, Tymoczko JL, Stryer L., Biochemistry, 5th edition Chapter 8 Enzymes: Basic Concepts and Kinetics. 8.4.2 (W H Freeman, 2002).

Reference 2: "The turnover number of an enzyme (k_{cat} or catalytic rate constant) is the maximal **number of molecules of substrate converted to product per active site per unit time** of several different substrates to different products."

Roskoski R., Michaelis-Menten Kinetics, Reference Module in Biomedical Sciences (2015).

DOI:10.1016/B978-0-12-801238-3.05143-6

b. The authors state in their rebuttal "We respectfully disagree with this point, because our Equations 3 and 4 are correct and follow the Michaelis-Menten Equation as it is described in biochemistry textbooks " and "The Michaelis-Menten Equation is valid also at low metabolite concentrations. The Michaelis-Menten Equation is the gold standard for ODE models of metabolic pathways and it is widely used in the modelling community (described for example here, doi: 10.1186/1742-4682-3-41)."

i. The point is not whether the Michaelis-Menten Equation (MME) is the "gold standard" (and it isn't in this case), but whether it is being used appropriately in the model. For the reasons listed below, I don't think it is appropriate.

ii. MME is not typically used to model degradation (MME in the production term is fine) because the removal of most chemical species is usually considered a first order reaction. If it is treated as zero order (which should be explained) it should still have units of nM min^{-1} of FBP. If the authors wish to use MME in Eq. 1 they need to explain why. One interpretation of Eq. 1 is that the authors are stating that the clearance of FBP by the enzymes is dependent on the function $\text{FBP}/(\text{FBP}+K)$. What does this mean? Because this is dimensionless, does this portion of the equation describe the probability of clearance for each molecule of FBP at a given concentration of FBP for each enzyme molecule. A single molecule of FBP will have a lower probability of clearance by an enzyme when the FBP concentration is low and a high probability of clearance

by an enzyme when the FBP concentration is high. This is the opposite of what is expected because at high concentrations you would expect competition between FBP molecules for the enzyme binding site. I recommend the reference in section 1d in this review for more information on the MME and Chapter 2 in *An Introduction to Systems Biology (Design Principles of Biological Circuits)* by **U. Alon** for modeling molecule clearance.

Point 1b is not correct: i) Even the Uri Alon group applies the exact same modeling format with Michaelis-Menten Equation to combine a metabolic pathway with enzyme expression (REF 3) and ii) we don't simulate a degradation process but enzyme catalyzed reactions.

The reviewer cites **work from the Uri Alon** group, and one of their models has the exact same structure as our: it combines metabolism and enzyme expression via the Michaelis Menten equation (REF 3):

Reference 3: “The enzymes E1, E2 and E3 produce products according to standard **Michaelis-Menten equations**, in which the rate of production, utilization and dilution of metabolite S_i is described by equation 1 in Box 1 in which α is the cell division rate. [...] The concentration of each enzyme is a balance of its expression rate and dilution by cell growth.”

Zaslaver, ..., Alon (2004) Just-in-time transcription program in metabolic pathways. *Nature Genetics* 36. <https://doi.org/10.1038/ng1348>

We disagree with the concern about degradation processes, because we simulate enzyme catalyzed reactions that converts FBP into another metabolite (which is not part of our model and outside the system boundaries).

The MME is one of the most established methods to simulate enzyme catalyzed reactions.

With MME, the FBP consuming reaction depends on the FBP concentration by the Michaelis Menten term $FBP/(FBP+K_m)$. This term ensures that if $FBP \gg K_m$, we get a zero-order reaction, because then the enzyme is saturated and operates at v_{max} . If $FBP \ll K_m$, the reaction rate depends linearly on the amount of FBP, and becomes a first-order reaction. This saturating behavior of the enzyme is an essential feature of enzymatic reactions and well described by Michaelis Menten kinetics.

Ref 4 – 7 are a few more important studies that used MME in exactly the same way as we did in our model:

Reference 4: “Converting the EM-based parameters into a **Michaelis–Menten** equivalent formalism revealed that 80% of K_m and k_{cat} parameters are within one order of magnitude of the literature-reported values.”

Khodayari,..., Maranas (2014) A kinetic model of *Escherichia coli* core metabolism satisfying multiple sets of mutant flux data. *Metab. Eng.* 25. <http://dx.doi.org/10.1016/j.ymben.2014.05.014>

Reference 5: “To demonstrate the model screening ability, we independently constructed a test model based on lumped **Michaelis-Menten-type kinetics** for each enzyme.”

Tran,..., Liao (2008) Ensemble Modeling of Metabolic Networks. *Biophys. J.* 95. <https://doi.org/10.1529/biophysj.108.135442>.

Reference 6: “To analyse the system quantitatively, we constructed a nonlinear ordinary differential equation (ODE) model on the basis of a dynamic balance of biomolecules. Enzyme kinetics were described by **Michaelis–Menten kinetics**.”

Fung,..., Liao (2005) A synthetic gene–metabolic oscillator. *Nature* 435. <https://doi.org/10.1038/nature03508>

Reference 7: “In this work, we compared the local transient response properties of dynamic models constructed using rate laws with varying levels of approximation. These approximate rate laws were: 1) a **Michaelis-Menten rate law with measured enzyme parameters**, 2) a Michaelis-Menten rate law with approximated parameters, using the convenience kinetics convention [...] We found that the **Michaelis-Menten** rate law with measured enzyme parameters yields an **excellent approximation** of the full system dynamics, while other assumptions cause greater discrepancies in system dynamic behavior.”

Du, B. et al. Evaluation of rate law approximations in bottom-up kinetic models of metabolism. *BMC Syst. Biol.* 10, 40 (2016). <https://doi.org/10.1186/s12918-016-0283-2>

iii. MME is usually used to simplify the modeling of chemical kinetics by making certain assumptions such as the total enzyme concentration is constant, and the substrate concentrations \gg enzyme concentration. However, the model is using MME without respecting these assumptions and that is potentially problematic. The authors declared that MME is valid at low metabolite concentration without providing a source. Here are two textbooks and even a Wikipedia entry with clear statements contradicting that declaration.

1. “That the substrate concentration is much larger than the enzyme concentration.” Page 43 in <https://link.springer.com/content/pdf/10.1007%2F978-1-349-01959-5.pdf>

2. “An essential feature of Michaelis–Menten kinetics is that the catalyst becomes saturated at high substrate concentrations.” <https://www.sciencedirect.com/topics/engineering/michaelis-menten-equation>

3. “Under certain assumptions – such as the enzyme concentration being much less than the substrate concentration – the rate of product formation is given by https://en.wikipedia.org/wiki/Michaelis%E2%80%93Menten_kinetics

Point 1biii is not correct: Michaelis Menten kinetics are also valid at low substrate concentrations. The statements above are out of context and do not refer to the assumptions underlying the Michaelis Menten equation:

Point 1biii 1): refers to conditions that are required to estimate kinetic parameters with *in vitro* enzymatic assays

Point 1biii 2): We don't understand why this is problematic. Saturation is an essential feature of enzymatic reactions and that's why we use Michaelis–Menten kinetics (and not other kinetic formats like power law, or lin-log kinetics)

Point 1biii 3): This sentence does not refer to assumptions of the Michealis Menten equation itself. It refers to the assumption when a zero order approximation is valid. This is not well written on Wikipedia, and perhaps not the best reference.

c. It is not clear why simple mass action kinetics was not used instead of MME because the predictions are simulated (as opposed to an analytical solution), and it would be very feasible to add a few more equations to the model. This approach is more standard because it explicitly models each reaction step and each species. If the model must be included in the model, why not explicitly model the enzyme-substrate complex and the degradation of the substrate? Then the model would be applicable for all the conditions that are being modeled, including dynamically varying enzyme concentration, and there would be no concern about any of the assumptions of the MME, including a constant total enzyme conc, product formation being slower than association and dissociation of the enzyme-substrate complex, and the enzyme conc being less than the substrate conc.

We disagree that mass action kinetics are more standard. Databases like BRENDA or EcoCyc report enzymatic parameters in the form of Michaelis Menten constants and not the constants of each micro step of an enzymatic reaction. A very important paper about *in vivo* parameters of enzymes makes this point (Ref 8, which has more than 1000 citations):

Reference 8: “Understanding the extent of saturation can allow one to reduce Michaelis-Menten equations into simpler expression, by omitting (taking as unity) terms for substrates that are reliably saturating, and by simplifying terms for substrates that are never saturating as first-order approximations. **This facilitates otherwise intractable parameter identification challenges** in building dynamic Michaelis-Menten models of metabolism.”

Bennett,..., Rabinowitz (2009) Absolute metabolite concentrations and implied enzyme active site occupancy in *Escherichia coli*. *Nature Chem. Biol.* 5. <https://doi.org/10.1038/nchembio.186>

d. I refer the authors to Chapter 6 (Modeling Molecular Interaction Networks with Nonlinear Ordinary Differential Equations by Emery D. Conrad and John J. Tyson) in System Modeling in Cellular Biology: From Concepts to Nuts and Bolts (Eds: Zoltan Szallasi, Jorg Stelling, and Vipul Periwa) on the application of MME and mass action kinetic models.

2. EMRA.

a. It is unclear that the experiments match the predictions for the 2xcra model. The axes on the plots on Fig. 3c (model) and 3f are different so it is not possible for the reader to directly compare them, and therefore to draw any conclusion as to whether the experiments do or do not agree with the predictions.

Our model is not supposed to make quantitative predictions. The model makes qualitative predictions about glycerol fluxes in the three models relative to each other. These predictions are:

1. The Δ cra model achieves higher glycerol fluxes than the base model (this confirms our hypothesis of misregulation by Cra)
2. The 2xcra model achieves the highest glycerol fluxes of all three models (which is an important design prediction that we confirmed experimentally)

b. The comment in the previous review on equilibrium vs steady state, was one of several general questions posed to prompt the authors to provide more details about the analysis. For example, have the authors considered cases where the eigenvalue had no zero real parts but a complex part, such as stable oscillations. This is an important consideration in a system with feedback. Following from the above, the manuscript states for the analysis “1st criterion: all real parts of the eigenvalues of the system’s Jacobian need to be negative.” “Once an eigenvalue reaches zero, the Jacobian becomes singular and matrix inversion is no longer possible. This bifurcation point defines the boundary between the stable and unstable parameter space. In other words: after this point is passed, the system can’t return to a stable steady state.” Should the cut-off be eigenvalues >0 to permit stable oscillations?

The suggested analysis is not possible, because the complexity of the model prevents an analytical solution. Therefore, we rely on a numerical method to determine eigenvalues. With this numerical method it is not possible to move past eigenvalues ≥ 0 , and therefore we cannot analyze sustained oscillations (oscillations with constant amplitudes). We also don’t understand why this analysis should be relevant for our conclusions.

In detail, due to the model complexity we rely on a numerical parameter continuation method instead of solving for the eigenvalues analytically. At each step of this parameter continuation we calculate the inverse of the Jacobian Matrix at each step. This is only possible as long as the eigenvalues of the Jacobian Matrix are negative (the matrix is regular). Thus we can only asymptotically approach zero and terminate the calculation when the biggest eigenvalue reaches a value of $-1E-5$. This is also explained in the original paper that describes the EMRA method (also with a model based on Michaelis-Menten kinetics):

Reference 9: “In practice, the Jacobian almost always becomes badly conditioned when the system is approaching a bifurcation point. We consider such an edge case a bifurcation point. Additionally, due to the nature of numerical integration, it is possible to “jump” over the region of singularity. To account for this, we routinely check if any of the eigenvalues of the Jacobian matrix has crossed the zero line to detect if the system has passed through a bifurcation point.”

Lee, ..., Liao (2014) Ensemble Modeling for Robustness Analysis in engineering non-native metabolic pathways. Metabolic Engineering 25. <https://doi.org/10.1016/j.ymben.2014.06.006>

c. Because the EMRA is based on the ODE model, and it has the issues described above in comment 1, I don't have confidence in the predictions of the EMRA model.

This is a circular argument. We thus refer back to our response to point 1.

Reviewer #2 (Remarks to the Author):

Summary

The authors made unsupported claims in previous revisions about ultrasensitivity, robustness and the role of feedback regulation in their system, which raised questions about their understanding of those topics as well as the modeling of biological and chemical systems. Given these concerns, the authors were asked to share more information about their models, modify them so the assumptions and processes were more transparent, and/or remove them. The authors have not been receptive to or responded adequately to some of this feedback over multiple review cycles. Therefore, it still remains unclear what assumptions are built into the models and whether they are appropriate given their purpose or parameter values. Importantly, there are also no quantitative or non-trivial qualitative predictions from the models that are validated by experiments. Without experimental results to test specific predictions from these models, they have no purpose and further discussion over their details is hypothetical and will likely continue to be unproductive.

Specific comments:

1) a. The units for kcat are now correct in the rebuttal letter. These units are not the same as in the previous and current versions of the manuscript.

b. i) As mentioned in the previous review, the key point is not whether an equation has been published before, but whether it is appropriate in this study.

b. ii) It is unclear why the authors rejected the reviewer's suggestion to include an additional term for first order degradation without an enzyme, and then cited at least two papers in their rebuttal which have done so. Please see equation 1 of box 1 in Zaslaver et al., which has a negative $\alpha \cdot S_i$ term for cell clearance, and the first equation in S1 of Fung et al. 2005, which has a $k_{TCA} \cdot \text{AcCoA}$ clearance term for the flux to ethanol production. At low enzyme concentrations and at some parameter values, these additional degradation terms will have a large impact on the behavior of the system. Further, in the absence of enzymes 1 and 2, FBP will accumulate without any upper bound if there is no term for FBP clearance that does not depend on an enzyme. This highlights again a general concern with the modeling by the authors, it is not rigorous and there are many assumptions that do not appear to have fully considered.

b. iii) The authors' response was not helpful in explaining why they believe that MME can be used at low substrate concentrations. The equation ($V_0 = k_{cat} \cdot E \cdot [S] / ([S] + K_m)$) is derived from modeling the concentration of ES complex. Often the free substrate concentration, $[S]$, is assumed to be equal to the initial substrate $[S_0]$ (in a closed system) or the total substrate concentration $[S_{tot}]$. These approximations are appropriate because $[S] \gg [ES]$, and therefore the small amount of substrate bound to ES can be ignored. However, when $[S] \ll [E]$ this is not true. In the FBP model, it is unclear whether the total FBP concentration or free FBP concentration is measured, and how the authors have accounted for the amount of substrate in the ES complex. For these reasons, MME is generally regarded as inaccurate at low substrate concentrations and not an appropriate when $[S] \ll [E]$. Derivation and assumptions can be found here:

<https://www.bgu.ac.il/~afialoc/BioHTML/Goodies/DeriveMMEqn.html>

c. See response to 1 b i). The authors response did not address all the questions or explain why they did not try all the reviewer's suggestions.

2) EMRA is based on the model above, therefore there cannot be any further progress on this topic until the concerns above are addressed. In the rebuttal the authors stated that EMRA is unable to make quantitative predictions and is only used to make a very basic qualitative prediction – the relative level of glycerol flux in three strains. This qualitative prediction does not need a model and can be determined from Fig. 3b. As stated in previous reviews, it is unclear how the reader should interpret the output of EMRA (the connection between Fig. 3c and Fig. 3e,f is not obvious), and the authors have not addressed this.

The authors have also confirmed that EMRA cannot evaluate stable oscillations, which is an important limitation. The reason for asking whether it can do so, is that oscillations can occur with feedback and metabolic systems, as illustrated by a paper cited by the authors in their rebuttal

(Ref 6 Fung et al, 2005). In summary, the authors have not shown EMRA adds value to study, that its output is supported by experimental results, or it is a suitable tool for this system.

Reviewer #3 (Remarks to the Author):

NCOMMS-20-29588C-Z Metabolome and proteome analyses of glycerol overproducing E. coli unravel transcriptional misregulation of glycolysis and guide optimization strategies

Summary of paper: aim and experimental results.

The authors express the yeast glycerol biosynthetic pathway in E. coli under an inducible promoter and demonstrate that the pathway's induction causes a strong growth defect. They identified that the cause of this defect is due to disruption of the E. coli natural gene regulatory network due to activation of the Cra transcription factor. In the glycerol producing strain this leads to downregulation of glycolytic enzymes and upregulation of gluconeogenesis enzymes. The authors demonstrate that in a Cra mutant strain this burden effect is reduced/abolished. The authors go on to redesign their inducible promoter so that it is repressed by Cra. They demonstrate that this hybrid promoter results in better glycerol yields and reduces the burden, and demonstrate through metabolomics and proteomics that this is through a dynamic repression by Cra. The authors also demonstrate a similar effect by weakening their inducible promoter. The authors show their hybrid inducible promoter can improve the growth of carotenoid over producing E. coli strains.

Summary of modelling

The authors take a two-pronged modelling approach: (1) using both flux balance analysis to assess the growth / glycerol trade off and (2) a small kinetic model of the F-bis-P branch point to study its regulatory structure. The cause of the impasse between authors and reviewers is regarding the kinetic model and the authors' approach is summarised below:

The authors consider a simple metabolic network. FBP is produced at rate $r_{\text{upperglycolysis}}$ and is converted to glycerol through the action of enzyme e_2 and enters lower glycolysis through enzyme e_1 . These reactions' rates are r_{glycerol} and $r_{\text{lowerglycolysis}}$ respectively. The authors model the action of e_i using Michaelis-Menten kinetics.

The authors model the gene expression of enzymes assuming that they are produced at rate β_i and dilute due to growth, μ . The authors model regulation of gene expression by scaling the maximum value of β_i by $(\text{FBP}/(\text{FBP}_{\text{ss}}))^{\alpha_i}$. To remove the impact of FBP regulation, α_i is set to zero. The authors estimate $\beta_{(1,\text{max})}$ from experimental proteome measurements and $\beta_{(2,\text{max})}$ from a simple assumption of maximal protein synthesis.

The authors dynamically model growth rate μ as $r_{\text{lowerglycolysis}}/((5747 \text{ mM}))$.

Initial review of this modelling approach

B1 The authors do not explicitly model the concentrations of either the lower glycolytic species or glycerol.

B2 The authors do not account for differing update takes across at different growth rates – or provide a reference as to why this can be assumed to be constant. The authors also do not say how they determine the rate of influx rate as 66.67 mM/min as it not clear what "2 uL per mg cell volume" corresponds to.

B3 I would have expected Hill functions not to be used for scaling β_i in response to FBP and the authors do not describe why they have chosen $(\text{FBP}/(\text{FBP}_{\text{ss}}))^{\alpha_i}$. The authors also don't explain their rationale for varying α_i between 1 and 2 only.

B4 It would be helpful if the authors named enzyme e_1 to aid replication. For example, while the authors obtain the concentration of e_1 from the dataset in Schmidt et al (2016), it is not clear

which enzyme this is. No growth rate is given, making it impossible to assess if the value of $\beta_{(1,max)}$ is reasonable. For example, is the growth rate similar to those seen experimentally in this work or is it 0.6 per h as used in the determination of $\beta_{(2,max)}$ (if so then $\beta_{(1,max)}$ should be 0.00023 mM per min, not 0.00027 mM per min)?

B5 The authors make a fair attempt to calculate the maximal value of $\beta_{(2,max)}$ based on the protein size and the number of translating ribosomes at a growth rate of 0.6 per h. They assume that at maximum induction all ribosomes translate the exogenous GPD1, this is contradicted in the observations by Scott et al (2010). In this work Scott et al estimated that at most half ribosomes can translate exogenous proteins and even this is associated with significant growth defects – these significant defects would likely lead to differing uptake rates (see point B2 above).

B6 Whilst I agree that the growth rate is likely proportional to the flux through lower glycolysis, the authors have not explained their rationale for this assumption. They do not explain why this constant of proportionality is 1/5747.

B7 The authors do not account for dilution of FBP by growth (a $-\mu \cdot \text{FBP}$ term). I assume the authors assume that the flux to lower glycolysis $r_{(lower_glycolysis)}$ dominates the dynamics? This would also be a reasonable assumption but I feel it should be stated. Do the authors think this assumption is fair even when e_1 levels are low?

B8 The authors do not refer to their robustness analysis in the main text and it is currently not clear why this has been carried out.

Summary of the Reviewer 2's criticisms of the modelling approach at the third revision

R1 Michaelis-Menten is not an appropriate modelling formalism because:

- a) The equation dimensions and units do not balance
- b) M-M is not used to model degradation which should be a first order equation of the form $-\alpha \cdot X$.
- c) This model violates the assumption that substrate concentrations are greater than enzyme concentrations needed for M-M.
- d) This model violates the assumption that the enzyme concentration is constant needed for M-M.
- e) The reviewer argues mass action kinetics should be used instead.

R2 The model predictions for 2xcra do not match the experimental results in Fig 3.

R3 In their ensemble model robustness analysis, the authors should consider more potential eigenvalue values, such as positive or complex numbers.

R4 The review has no confidence in the EMRA given their criticism of the underlying model.

Summary of author rebuttal and appeal

A1 The authors give a detailed discussion of the units of k_{cat} and successfully demonstrate that their units are consistent.

A2 In response to R1b, the authors argue that their model is not attempting to capture FBP dilution but rather FBP conversion by an enzyme to another metabolite. (This response also answers my comment B7 but as I say I think this should be more clearly stated).

A3 The authors re-assert that M-M kinetics are valid for low substrate concentrations.

A4 The authors argue that the model makes qualitative relative predictions and therefore does replicate the experimental data.

A5 The authors cite the original EMRA paper which notes the method identifies the boundary of instability; that is, it identifies the point at which the Jacobian matrix becomes singular and hence eigenvalues cannot be calculated.

Final conclusions

The authors use an ensemble modelling approach to study three topologies; the natural case, the Δ cra case (with no FBP regulation) and the dual 2xcra case (where FBP activates both glycolysis and the pathway of interest). This use of an ensemble approach means that the precise parameters of the system are not needed. Instead of making quantitative predictions the authors approach allows qualitative insights. The authors' models predict that's both Δ cra and 2xcra show potential higher rates of glycerol production over the base case (Fig 3c). An in vivo implementation demonstrates that this is the case (Fig 3e). I disagree with comment R2.

I have carefully reviewed the authors' model derivation as outlined above and any concerns I have are summarised in comments B1-8. As you will see the majority of my comments relate to the authors communication. Whilst I do not think at present all of the authors' assumptions are well explained, I do not agree with the previous reviewer's criticisms of their approach.

Regarding the comments R1c (low molecule concentration) and R1d (enzyme conservation) I have two thoughts. Firstly, if molecule concentrations are so low that M-M cannot apply, then an ODE framework would be inappropriate and a stochastic method would be required, and secondly it would be fair to argue that as metabolic processes happen on the order of seconds and gene expression takes minutes then for the purposes of modelling metabolic rates the enzyme concentration is approximately constant. The authors could show this by expanding their mechanism and doing the necessary time scale separation analysis.

At present the authors did not discuss their robustness analysis in the main text. They do not discuss what instability may mean in the context of their system; for example do they mean their system will oscillate or grow exponentially? They do not discuss the consequences for biological implementation of these different instabilities.

In conclusion, with a more complete description of the underlying assumptions as outlined in my queries B1-8 I do not see any fundamental issues with the modelling approach which would preclude publication.

Additional comments on the rest of the manuscript

Figure 1d, 2e and 3e OD curves lack a legend. I also note that the pinkest lines (which I assume these are 2% arabinose) do not reach exponential or stationary phase in 20 h. It may be worth noting in the main text that the data for high arabinose is not from mid-exponential phase cells.

REVIEWER COMMENTS

Reviewer #3 (Remarks to the Author):

NCOMMS-20-29588C-Z Metabolome and proteome analyses of glycerol overproducing E. coli unravel transcriptional misregulation of glycolysis and guide optimization strategies

Summary of paper: aim and experimental results.

The authors express the yeast glycerol biosynthetic pathway in E. coli under an inducible promoter and demonstrate that the pathway's induction causes a strong growth defect. They identified that the cause of this defect is due to disruption of the E. coli natural gene regulatory network due to activation of the Cra transcription factor. In the glycerol producing strain this leads to downregulation of glycolytic enzymes and upregulation of gluconeogenesis enzymes. The authors demonstrate that in a Cra mutant strain this burden effect is reduced/abolished. The authors go on to redesign their inducible promoter so that it is repressed by Cra. They demonstrate that this hybrid promoter results in better glycerol yields and reduces the burden, and demonstrate through metabolomics and proteomics that this is through a dynamic repression by Cra. The authors also demonstrate a similar effect by weakening their inducible promoter. The authors show their hybrid inducible promoter can improve the growth of carotenoid over producing E. coli strains.

Summary of modelling

The authors take a two-pronged modelling approach: (1) using both flux balance analysis to assess the growth / glycerol trade off and (2) a small kinetic model of the F-bis-P branch point to study its regulatory structure. The cause of the impasse between authors and reviewers is regarding the kinetic model and the authors' approach is summarised below:

The authors consider a simple metabolic network. FBP is produced at rate $r_{\text{upperglycolysis}}$ and is converted to glycerol through the action of enzyme e_2 and enters lower glycolysis through enzyme e_1 . These reactions' rates are r_{glycerol} and $r_{\text{lowerglycolysis}}$ respectively. The authors model the action of e_i using Michaelis-Menten kinetics.

The authors model the gene expression of enzymes assuming that they are produced at rate β_i and dilute due to growth, μ . The authors model regulation of gene expression by scaling the maximum value of β_i by $(\text{FBP}/(\text{FBP}_{ss}))^{\alpha_i}$. To remove the impact of FBP regulation, α_i is set to zero. The authors estimate $\beta_{(1,\text{max})}$ from experimental proteome measurements and $\beta_{(2,\text{max})}$ from a simple assumption of maximal protein synthesis.

The authors dynamically model growth rate μ as $r_{\text{lowerglycolysis}}/(5747 \text{ mM})$.

We thank the third reviewer for a thorough and constructive review of our manuscript, and for commenting on the points of reviewer 2.

Initial review of this modelling approach

B1 The authors do not explicitly model the concentrations of either the lower glycolytic species or glycerol.

This is correct, our model boundaries do not include the concentrations of glycerol and lower glycolysis metabolites. We did not include glycerol in the model analysis, because glycerol accumulates with a rate that is equal to $r_{glycerol}$, and thus glycerol levels are not at steady state (what complicates the robustness analysis). Lower glycolysis metabolites were excluded to keep the model as simple as possible under the assumption that FBP is directly consumed for growth. **We highlighted the model boundaries in Fig. 3a and show that FBP is either converted to glycerol or biomass.**

B2 The authors do not account for differing update takes across at different growth rates – or provide a reference as to why this can be assumed to be constant. The authors also do not say how they determine the rate of influx rate as 66.67 mM/min as it not clear what "2 uL per mg cell volume" corresponds to.

We used a constant glucose uptake rate in our model, because we assume that glucose is either used for growth or glycerol production. This means we assume the same mass balance that we used for the FBA analysis in Fig. 1e. This is now better explained in the revised text:

“Similar to flux balance analysis (Fig. 1e), we assumed a constant influx in the model and fixed the reaction rate in upper glycolysis to 4.9 mmol g⁻¹h⁻¹. This means that FBP is produced at a constant rate and it is either used for glycerol production or for growth according to the mass balances in Equation 1.”

The specific cell volume of 2 μL per mg cell dry weight was used to convert flux from 8 mmol g⁻¹h⁻¹ into 66.67 mM/min and we added the equation in the revised methods part. The value 2 μL per mg cell dry weight is from Bennett et al. 2008, and we added this as Ref 49 in the text. In the previous version of the model we assumed that all glucose goes into glycolysis, but a significant fraction goes into the pentose phosphate pathway. Thus, we revised the model and used FBA to calculate the upper glycolytic flux, which is **4.9 mmol g⁻¹h⁻¹**. We explain these calculations in the revised manuscript:

„An upper glycolytic flux of 4.904 mmol g⁻¹h⁻¹ was estimated with FBA using a glucose uptake rate of 8 mmol g⁻¹ h⁻¹. With the specific cell volume for E. coli (2 μl mg⁻¹)⁴⁹ the reaction rate $r_{upper\ glycolysis}$ is:

$$r_{upper_glycolysis} = \frac{4.904 \text{ mmol g}^{-1}\text{h}^{-1}}{0.002 \text{ l g}^{-1}} * \frac{\text{h}}{60 \text{ min}} = 40.87 \text{ mM min}^{-1} \quad \text{“}$$

B3 I would have expected Hill functions not to be used for scaling β_i in response to FBP and the authors do not describe why they have chosen $(\text{FBP}/\text{FBP}_{ss})^{\alpha_i}$. The authors also don't explain their rationale for varying α_i between 1 and 2 only.

The power law format allows us to use the same 5000 parameter sets for all three models, because at steady state and zero induction the power-law term $(\text{FBP}/\text{FBP}_{ss})^\alpha$ equals one. Hill equations would require different parameter set for each of the three models. We added this explanation at several points in the revised manuscript:

Results

“We simulated Cra-regulation with a power law term that affects the maximal enzyme expression rate. Since the power-law term equals one in the un-induced state, all models share the same parameter set.”

Methods

„Cra-regulation was simulated with a power law term $\left(\frac{\text{FBP}}{\text{FBP}_{ss}}\right)^\alpha$ that affects the maximal enzyme expression rate. The power-law format has the advantage that the power-law term equals one in the un-induced state and therefore allows the same parameter values for the base model, the Δ cra model and the 2x cra model. Further, setting α to zero removes the regulation and therefore α_2 was zero in the base model, while α_1 and α_2 were zero in the Δ cra model.“

We sampled the exponent (α_i) between 1 and 2, because with the lower bound of 1 we ensure that the expression rate is at least linearly dependent on the FBP concentration. The upper bound of 2 avoids higher-order dynamics that can cause instabilities. We clarified this in the method section of the revised manuscript:

“The power-law exponents, α_1 and α_2 , were randomly sampled between 1 and 2. The lower bound was 1 to ensure that the expression rate is at least linearly dependent on the FBP concentration. The upper bound was 2 to avoid higher-order dynamics that can cause instabilities⁵¹.”

B4 It would be helpful if the authors named enzyme e1 to aid replication. For example, while the authors obtain the concentration of e1 from the dataset in Schmidt et al (2016), it is not clear which enzyme this is. No growth rate is given, making it impossible to assess if the value of $\beta_{(1,\text{max})}$ is reasonable. For example, is the growth rate similar to those seen experimentally in this work or is it 0.6 per h as used in the determination of $\beta_{(2,\text{max})}$ (if so then $\beta_{(1,\text{max})}$ should be 0.00023 mM per min, not 0.00027 mM per min)?

The concentration of e1 is the concentration of glyceraldehyde-3-phosphate dehydrogenase (GapA) in *E. coli* MG1655 in the glucose condition of the Schmidt *et al.* data set. We selected GapA because it was the most sensitive glycolytic enzyme in the proteome data (Fig. 2d). We added the name of e1 in the revised manuscript

„Enzyme e1 corresponds to glyceraldehyde-3-phosphate dehydrogenase (GapA) in lower glycolysis and e2 is GPD1 in the glycerol pathway.“

The reviewer is correct that the growth rate was not fully consistent in the model:

- 0.69 per h was used to estimate $\beta(1,max)$ and it was the growth rate in the un-induced state. We selected the growth rate 0.69 per h because it followed from FBA analysis (see Fig. 1e).
- 0.6 per h was the growth rate at which the Bionumber data base reported the parameters that we used to estimate $\beta(2,max)$

We changed this and use now a consistent growth rate of 0.6 per h in the model analysis of the revised manuscript.

B5 The authors make a fair attempt to calculate the maximal value of $\beta(2,max)$ based on the protein size and the number of translating ribosomes at a growth rate of 0.6 per h. They assume that at maximum induction all ribosomes translate the exogenous GPD1, this is contradicted in the observations by Scott et al (2010). In this work Scott et al estimated that at most half ribosomes can translate exogenous proteins and even this is associated with significant growth defects – these significant defects would likely lead to differing uptake rates (see point B2 above).

We agree that the estimate of $\beta(2,max)$ was probably too high, because not 100% of the ribosomes can synthesize GPD1. Thus, we reduced the maximal fraction of GPD1 translating ribosomes to 20%. The model results with 20% are similar as with the initial 100%. We added this at different points in the revised manuscript and show the results of the new model analysis in Fig. 3:

“The fraction of ribosomes (p) that synthesize GPD1 at full induction was assumed to be 20%, because only 50% of the ribosomes can translate a heterologous protein and this is associated with significant protein burden⁵⁵. “

„... the continuation method terminates if it reaches the maximal expression rate of e_2 ($\beta(2,max)$), which we defined as the rate were 20% of the ribosomes translate e_2 .“

B6 Whilst I agree that the growth rate is likely proportional to the flux through lower glycolysis, the authors have not explained their rationale for this assumption. They do not explain why this constant of proportionality is 1/5747.

We assume a proportionality between the lower glycolytic flux and the growth rate for two reasons:

- The FBA results show a linear relationship between growth rate on the flux in lower glycolysis and we added this as a **new Supplementary Figure 11**.
- Previous work based on 13C flux analysis (Gerosa et al. 2015, *Cell Systems*) reports a lower glycolytic flux in conditions with lower growth rates.

The constant 1/5747 is a scaling factor to convert lower glycolytic flux (mM/s) into the specific growth rate (1/h). With the new growth rate of 0.6 per h the scaling factor changed to 1/4086. We clarify these points in the revised text:

*“We assumed that the growth rate μ is proportional to $r_{lower_glycolysis}$ because flux balance analysis showed a linear relationship between lower glycolytic flux and the growth rate (Supplementary Figure 11). Additionally, previous 13C labelling data showed a positive correlation between lower glycolytic flux and growth in *E. coli*⁵⁰. With a growth rate of 0.01 min^{-1} in the un-induced state, the proportionality factor follows as:*

$$\alpha = \frac{r_{lower_glycolysis}}{\mu} = \frac{40.86 \text{ mM min}^{-1}}{0.01 \text{ min}^{-1}} = 4086 \text{ mM} \quad (\text{Equation 9})$$

B7 The authors do not account for dilution of FBP by growth (a $-\mu \cdot \text{FBP}$ term). I assume the authors assume that the flux to lower glycolysis $r_{\text{(lower_glycolysis)}}$ dominates the dynamics? This would also be a reasonable assumption but I feel it should be stated.

That's correct, we did not include growth dilution of FBP in the model, because this term is much smaller than the flux in lower glycolysis and thus hardly affects FBP dynamics. Nevertheless, we revised the model and added a term for growth dilution of FBP to the model (as expected this hardly affected the results).

Do the authors think this assumption is fair even when e_1 levels are low?

We think the assumption is true even at low e_1 levels, because growth is proportional to $r_{\text{(lower_glycolysis)}}$ and thus it will always dominate dilution by growth (by the scaling factor mentioned in point B6).

B8 The authors do not refer to their robustness analysis in the main text and it is currently not clear why this has been carried out.

We used the continuation method to gradually increase the induction and to estimate the maximal glycerol production rate. The only purpose of the robustness analysis is to test if the model is stable at every iteration, because a high glycerol production rate in an unstable model is not meaningful. We tried to explain this better in the revised manuscript:

„To estimate $r_{\text{glycerol,MAX}}$ we made use of a numerical continuation method²⁴, which iteratively increases the expression rate of enzyme e_2 (b_2) and computes the new steady state for FBP, e_1 and e_2 . After each iteration, the continuation method determines the stability of the model by inspecting the eigenvalues of the Jacobian matrix²⁴, and terminates if instabilities occur in the model. If the model remains stable, the continuation method terminates at the maximal expression rate of e_2 ($\beta_{2,max}$), which we defined as the rate were 20% of the ribosomes translate e_2 . Thus, $r_{\text{glycerol,MAX}}$ is the glycerol production at the termination point of the continuation method and we obtained 5000 values of $r_{\text{glycerol,MAX}}$ for each of the 3 models (Fig. 3c). „

To better illustrate the concept, we simulated the models in the time domain to show the reason for instabilities and compare the results in the time domain with the continuation method (see new Supplementary Figure 5, and our answer below).

Summary of the Reviewer 2's criticisms of the modelling approach at the third revision

R1 Michaelis-Menten is not an appropriate modelling formalism because:

- a) The equation dimensions and units do not balance
- b) M-M is not used to model degradation which should be a first order equation of the form $-\alpha \cdot X$.
- c) This model violates the assumption that substrate concentrations are greater than enzyme concentrations needed for M-M.
- d) This model violates the assumption that that the enzyme concentration is constant needed for M-M.
- e) The reviewer argues mass action kinetics should be used instead.

R2 The model predictions for 2xcra do not match the experimental results in Fig 3.
R3 In their ensemble model robustness analysis, the authors should consider more potential eigenvalue values, such as positive or complex numbers.

R4 The review has no confidence in the EMRA given their criticism of the underlying model.

Summary of author rebuttal and appeal

A1 The authors give a detailed discussion of the units of k_{cat} and successfully demonstrate that their units are consistent.

A2 In response to R1b, the authors argue that their model is not attempting to capture FBP dilution but rather FBP conversion by an enzyme to another metabolite. (This response also answers my comment B7 but as I say I think this should be more clearly stated).

A3 The authors re-assert that M-M kinetics are valid for low substrate concentrations.

A4 The authors argue that the model makes qualitative relative predictions and therefore does replicate the experimental data.

A5 The authors cite the original EMRA paper which notes the method identifies the boundary of instability; that is, it identifies the point at which the Jacobian matrix becomes singular and hence eigenvalues cannot be calculated.

Final conclusions

The authors use an ensemble modelling approach to study three topologies; the natural case, the Δ cra case (with no FBP regulation) and the dual 2xcra case (where FBP activates both glycolysis and the pathway of interest). This use of an ensemble approach means that the precise parameters of the system are not needed. Instead of making quantitative predictions the authors' approach allows qualitative insights. The authors' models predict that both Δ cra and 2xcra show potential higher rates of glycerol production over the base case (Fig 3c). An in vivo implementation demonstrates that this is the case (Fig 3e). I disagree with comment R2.

I have carefully reviewed the authors' model derivation as outlined above and any concerns I have are summarised in comments B1-8. As you will see the majority of my comments relate to the authors' communication. Whilst I do not think at present all of the authors' assumptions are well explained, I do not agree with the previous reviewer's criticisms of their approach.

Regarding the comments R1c (low molecule concentration) and R1d (enzyme conservation) I have two thoughts. Firstly, if molecule concentrations are so low that M-M cannot apply, then an ODE framework would be inappropriate and a stochastic method would be required, and secondly it would be fair to argue that as metabolic processes happen on the order of seconds and gene expression takes minutes then for the purposes of modelling metabolic rates the enzyme

concentration is approximately constant. The authors could show this by expanding their mechanism and doing the necessary time scale separation analysis.

We agree with these comments, especially with the comments about enzyme conservation. The different time-scales between metabolic reactions and gene expression follow directly from the maximal reaction and maximal expression rates: for the average parameter set the difference of the lower glycolysis reaction rate and expression rate is more than 5 orders of magnitude: $v_{max,1} = e1 \cdot k_{cat,1} = 99.95 \text{ mM min}^{-1}$ and $b1 = 0.000238 \text{ mM min}^{-1}$.

At present the authors did not discuss their robustness analysis in the main text. They do not discuss what instability may mean in the context of their system; for example do they mean their system will oscillate or grow exponentially? They do not discuss the consequences for biological implementation of these different instabilities.

We agree that it was not clear what causes the instabilities. To better understand the nature of the instabilities, we simulated induction of the three models in the time domain. Therefore, we used the average parameter set, i.e. each parameter was set to the median value of the respective interval. Notably, the results that we obtain with the time-course simulation matched the results with the continuation method and show that instabilities occur because $e2$ accumulates. These results are shown in a new **Supplementary Figure 5** and we discuss them now in the main text:

“To better understand the origin of the instabilities, we performed time-course simulations with the three models and an average parameter set (Supplementary Figure 5). The time-course simulations matched the results obtained with the continuation method, thus confirming that both numerical approaches yield the same results. We simulated the models at different induction levels and the base model was not stable at higher induction, because enzyme $e2$ increased exponentially at a critical induction level. Thus, there is a critical point where the expression rate of $e2$ exceeds its dilution by growth. These imbalances are probably amplified by Cra -regulation, because Cra downregulates $e1$ and thus growth. „

In conclusion, with a more complete description of the underlying assumptions as outlined in my queries B1-8 I do not see any fundamental issues with the modelling approach which would preclude publication.

Additional comments on the rest of the manuscript

Figure 1d, 2e and 3e OD curves lack a legend. I also note that the pink lines (which I assume these are 2% arabinose) do not reach exponential or stationary phase in 20 h. It may be worth noting in the main text that the data for high arabinose is not from mid-exponential phase cells.

We added the legend to these figures.

Reviewers' Comments:

Reviewer #3:

Remarks to the Author:

NCOMMS-20-29588C-Z Metabolome and proteome analyses of glycerol overproducing E. coli unravel transcriptional misregulation of glycolysis and guide optimization strategies

We were approached to review the modelling work done in this study. We saw no significant problems on our first review and we feel that the authors' most recent updates have greatly improved the communication of their approach. We think the authors have strengthened their work by including dilution of all species which will remove any concerns that the original assumption of no dilution for FBP may have caused readers. Whilst we suspect many other modellers would have chosen to use a hill function formulation for gene expression activation/inhibition, the power law format is an acceptable alternative, and the authors now describe why they have chosen this allowing the reader to make their own judgement. The authors' clarifications of the origins of some of their constants or assumptions about the linearity of growth rate are now well referenced allowing future replication of this work. The new text on the robustness analysis demonstrates its use within the manuscript and the short exploration of time course simulations begins to identify the potential cause of instabilities. Whilst this manuscript lacks a full analysis of the cause of instability between the three network topologies, given the authors' focus approach is to "design-and-build" we think such an analysis would probably not be within scope. We understand why the authors have an upper bound of two for their power law exponent's as they are right that introducing a significant nonlinearity would obviously lead to instabilities. We will not ask the authors to repeat their analysis with a higher upper bound, but we ask that they insert a sentence in the main text, in addition to the sentence in the methods, stating that exponents were only sampled between one and two. This will allow the reader to bear this fact in mind as they read the subsequent analysis and will prevent any reader over-interpreting the benefits of the 2xcra topology, as this topology may not be robust if the regulator was engineered with a higher cooperativity.

REVIEWER COMMENTS

Reviewer #3 (Remarks to the Author):

NCOMMS-20-29588C-Z Metabolome and proteome analyses of glycerol overproducing E. coli unravel transcriptional misregulation of glycolysis and guide optimization strategies

We were approached to review the modelling work done in this study. We saw no significant problems on our first review and we feel that the authors' most recent updates have greatly improved the communication of their approach. We think the authors have strengthened their work by including dilution of all species which will remove any concerns that that the original assumption of no dilution for FBP may have caused readers. Whilst we suspect many other modellers would have chosen to use a hill function formulation for gene expression activation/inhibition, the power law format is an acceptable alternative, and the authors now describe why they have chosen this allowing the reader to make their own judgement. The authors' clarifications of the origins of some of their constants or assumptions about the linearity of growth rate are now well referenced allowing future replication of this work. The new text on the robustness analysis demonstrates its use within the manuscript and the short exploration of time course simulations begins to identify the potential cause of instabilities. Whilst this manuscript lacks a full analysis of the cause of instability between the three network topologies, given the authors' focus approach is to "design-and-build" we think such an analysis would probably not be within scope. We understand why the authors have an upper bound of two for their power law exponent's as they are right that introducing a significant nonlinearity would obviously lead to instabilities. We will not ask the authors to repeat their analysis with a higher upper bound, but we ask that they insert a sentence in the main text, in addition to the sentence in the methods, stating that exponents were only sampled between one and two. This will allow the reader to bear this fact in mind as they read the subsequent analysis and will prevent any reader over-interpreting the benefits of the 2xcra topology, as this topology may not be robust if the regulator was engineered with a higher cooperativity.

We thank the reviewer for the helpful and positive comments. We included a sentence in the main text that states that we sampled exponents between 1 and 2:

"We sampled the power-law exponent between 1 and 2, in order to ensure that Cra-regulation depends at least linearly on the concentration of FBP and to avoid instabilities that can occur at exponents greater than 2."